# Land use and land cover changes along the China-Myanmar Oil and Gas pipelines – Monitoring infrastructure development in remote conflict-prone regions

Thiri Shwesin Aung[1]*, Thomas B. Fischer[2,3]ᵒ, John Buchanan[1]ᵒ

**1** Faculty of Arts and Sciences, Asia Center, Harvard University, Cambridge, MA, United States of America, **2** Environmental Assessment and Management Research Centre, Geography and Planning, School of Environmental Sciences, University of Liverpool, Liverpool, United Kingdom, **3** Research Unit for Environmental Science and Management, Faculty of Natural and Agricultural Sciences, North West University (Potchefstroom Campus), Potchefstroom, South Africa

ᵒ These authors contributed equally to this work.
* thirishwesinaung@fas.harvard.edu

**Data Availability Statement:** All relevant data are within the manuscript and its Supporting Information files.

**Funding:** Unfunded Studies.

## Abstract

Energy infrastructures can have negative impacts on the environment. In remote and / or sparsely populated as well as in conflict-prone regions, these can be difficult to assess, in particular when they are of a large scale. Analyzing land use and land cover changes can be an important initial step towards establishing the quantity and quality of impacts. Drawing from very-high-resolution-multi-temporal-satellite-imagery, this paper reports on a study which employed the Random Forest Classifier and Land Change Modeler to derive detailed information of the spatial patterns and temporal variations of land-use and land-cover changes resulting from the China-Myanmar Oil and Gas Pipelines in Ann township in Myanmar's Rakhine State of Myanmar. Deforestation and afforestation conversion processes during pre- and post-construction periods (2010 to 2012) are compared. Whilst substantial forest areas were lost along the pipelines, this is only part of the story, as afforestation has also happened in parallel. However, afforestation areas can be of a lower value, and in order to be able to take quality of forests into account, it is of crucial importance to accompany satellite-imagery based techniques with field observation. Findings have important implications for future infrastructure development projects in conflict-affected regions in Myanmar and elsewhere.

## Introduction

Energy development has various potential environmental and social challenges. While coal, oil and gas generation have been the primary focus of research on how landscapes may change, pipeline operations have received less attention. However, over recent years, oil and gas pipelines have undergone substantial expansion globally [1]. This is connected with the quest for

**Competing interests:** No authors have competing interests.

energy security and the development of reliable transnational energy sources. In Asia, particularly in China and India, energy and the associated development of infrastructure is an important driving force which also has an impact on politics and governance regimes.

With demand for energy growing very rapidly in China and India, Myanmar, sandwiched between them, has become both, a significant energy source and a transit corridor for energy, in particular fossil fuels. As a result, deals on oil and gas exploration and transmission pipelines are made between Myanmar and the two countries [2]. The Shwe gas project and the Myanmar-China oil transport project, commonly referred to as the "Myanmar-China Oil and Gas Pipelines," are two of the largest and most prominent energy projects in Myanmar, including the production and transportation of petroleum and natural gas located off the coast of Myanmar's Rakhine State. The pipeline project began in 2004 as a tri-nation energy collaboration to transport natural gas from offshore platforms in the Bay of Bengal off Rakhine State to India. However, in 2006, Myanmar signed an agreement with China to transport gas from Myanmar's offshore blocks and oil from Africa and the Middle East through overland pipelines to China's Yunnan Province [3].

While offering multiple economic benefits, there are many environmental (along with social) concerns about oil and gas pipelines as they can affect forest, farmland, and residential areas during construction and operation [4]. For example, in Pennsylvania, the impact of natural gas pipelines on forest areas were found to substantially exceed the impacts of all other energy development types (Johnson et al., 2011). In the Niger Delta, construction of oil and gas pipelines came with approximately 495 ha of forests being cleared, and nearly 10M trees being destroyed [5]. For Myanmar, [6] was reported that the Yadana-Yetagun pipeline in southern Burma has had serious environmental impacts. Inadequate environmental planning and negligence of environmental protection during the construction and operation stages were also documented [6, 7].The China-Myanmar oil and gas pipelines run across the Rakhine Roma Mountain Range, central Burma, and Shan State, traversing diverse ecosystems, dense forests, and rivers.

While both, the Myanmar and Chinese governments believe that oil and gas extraction and pipeline projects have the potential to benefit both nations substantially, contributing to significant economic opportunities, there are some grave concerns over their impacts on forests and cultivated lands, as well as on water and wildlife. This can significantly affect the livelihood of local communities [8]. The Rakhine Roma mountain range is recognized as one of the ten most vulnerable ecoregions in the world, feeding two crucial watersheds, the Brahmaputra and Irrawaddy Rivers, upon which many people depend for their livelihoods [9]. Although pipelines are usually buried underground, their construction, maintenance and monitoring require extensive clearing of land. Pipeline Right-Of-Way (ROW) often results in significant and permanent severance and fragmentation of forests and other natural habitats [10]. Pipeline ROWs are strips of land in which pipelines are located and which managing companies have legal rights to access. In Myanmar, pipelines cross Rakhine State, which has witnessed communal and militarized violence. They then pass through various conflict-affected regions in Shan State [11]. There are, therefore, numerous security concerns.

Environmental governance of large-scale infrastructure projects in this region is often criticized for being weak and projects in Myanmar have been associated with wide-ranging ecological destruction and human rights violations [12]. The pipelines have the potential to leave an extensive spatial footprint across Myanmar. There are ongoing civil conflicts in Rakhine and environmental problems associated with energy projects have the potential to exacerbate them [13]. Human rights issues and negative environmental impacts of pipeline development have been recorded in western and central Myanmar and there is evidence of land confiscations without compensation, forced relocations, damage of farming and community lands,

and destruction of natural hydrological conditions [14]. In Rakhine State alone, 1824 acres of agricultural land have been destroyed. Overall, clearing, drilling, and construction of pipelines and the construction and management of related infrastructures had detrimental effects on local environments and communities, especially in environmentally sensitive regions. It has also been documented that much of the adverse impacts of the project have affected Rakhine State, where environmental and social impacts are the most pronounced [15]. Importantly, there were no environmental regulations—including legally mandated environmental impact assessments (EIA)–at the time of the planning and development of the projects. This has been said to have accelerated negative consequences [16]. Although–rudimentary–EIAs are now undertaken, there has been no public participation during the process, with EIA reports not being publicly available [17].

Motivated by these concerns, in this paper the authors seek to address critical issues, such as the rate and pattern of LULCC along the pipelines, the extent of forest loss during the study period and the pattern of afforestation in the study area. Adverse impacts of pipelines on Land-Use-Land-Cover (LULC), with a particular focus on forest cover, and the pattern of Land-Use-Land-Cover-Change (LULCC) are reported on.

Myanmar possesses some of the largest remaining forest areas in Asia. Bhagwat et al. (2017) found that 63% of Myanmar was covered by forests in 2014. Nevertheless, the country is suffering from significant annual forest loss due to infrastructure development, firewood over-exploitation, illegal logging, shifting cultivation, and an expansion of agricultural lands [18]. [19] documented that a substantial increase in foreign investments, natural resource exploitation and land confiscation during civil wars were major underlying drivers of forest degradation in Myanmar. Increased commercial agriculture concession has also led to forest loss [19, 20]. Infrastructure and energy development have been identified as one of the most critical issues likely to affect Myanmar's forests [21].

in this context, the country has been said to suffer from limited institutional capacity to deal with these issues [22]. According to the FAO, forest cover declined from 41.196 million ha (61%) to 29.388 million ha (43%) between 1975 and 2015 [23]. This equates to a total decline of 11.8 million ha of forests in 70 years. Annually, forest coverage has declined by 0.3% during 1990s and 2000s [24].and during 1988 to 2017, the annual deforestation rate was reported as being 0.87% [25], the difference being explained by afforestation efforts. The impact of increased energy development pressures on forests and other land cover types, as well as wildlife, are largely unknown and undocumented.

## Study region

The study region is located along the China-Myanmar Oil and Gas Pipelines in Ann township of Kyaukpyu District in Myanmar's western-most state of Rakhine (See Fig 1). It has a tropical monsoon climate, featuring warm temperatures throughout the year and high annual rainfall with most of the rainfall from June to August. The climate is dominated by the Northeast and Southwest monsoons, and annual rainfall shows an increasing trend within the 1981–2018 baseline period. The township receives an average annual rainfall of about 4655mm, one of the highest average annual rainfalls in Myanmar [26]. Due to its physiography and climate patterns, Kyaukpyu has one of the densest forestlands and the most extensive areas of endangered biodiversity in Myanmar. The majority of the region is covered by forests, agricultural land, and tidal floodplains. Most of the primary forests found in the hilly areas contain some of the most ecologically significant habitats in the region [27]. The area exhibits a diverse terrestrial flora, including mangrove, shrubland, woodland, grassland, and several terrestrial, inland wildlife and reptile species. 36.56% (14,8527.49 acres) of the total district area (434,144 acres)

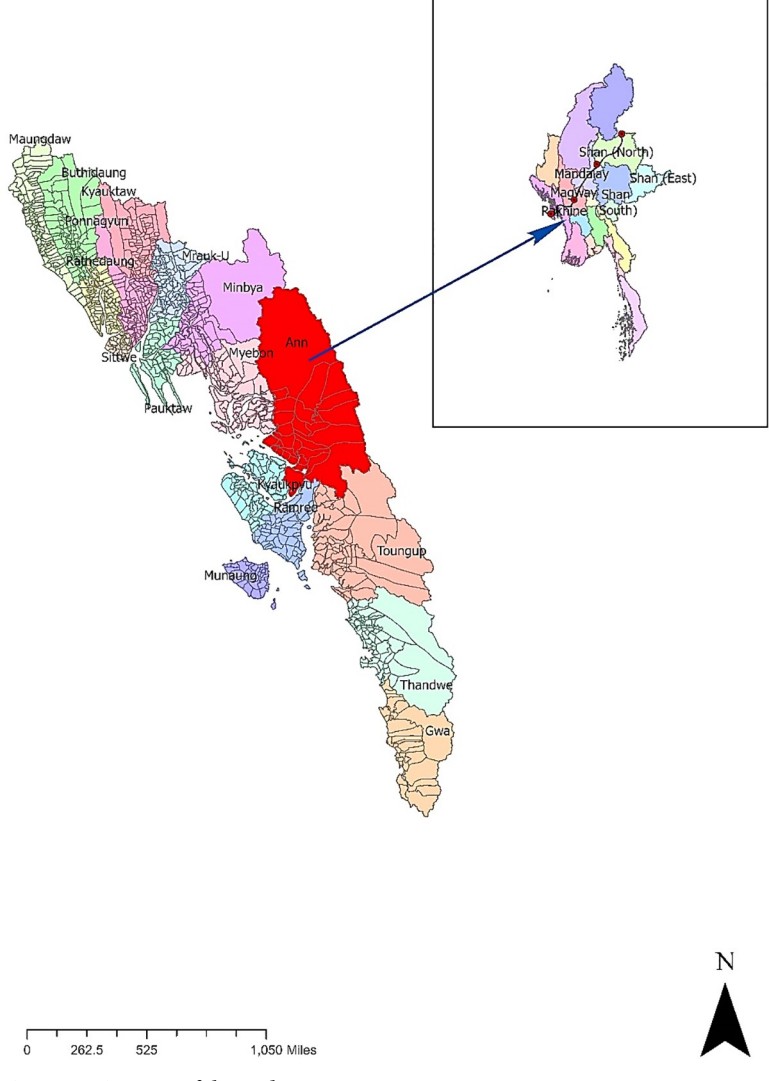

**Fig 1. Location map of the study area.**

is covered by forested areas. However, this number has decreased over time due to extensive anthropogenic activities, such as infrastructure and industrial development, and the construction of a Special Economic Zone (SEZ) [27]. The total population in Ann Township is 119,714. Although Rakhine State is endowed with an abundance of natural resources and is in a strategic location, it is one of the least economically developed areas in Myanmar. The majority of the population depends on agriculture and fishing for their livelihood. Overall, the quality of life for much of the population is considered low, especially in rural areas, given poor access to health and education services, inadequate infrastructure, low employment rates and income.

Land use in the Kyaukpyu District consists of forested and cultivated land, scrubland, non-forested land and cultivable wasteland, as well as protected land area. Some of the land has never been used for cultivation and may or may not be covered by forests. This type of land occupies approximately 53% of the region. Rural, urban, and industrial land use constitutes only 1% of total land use. Over 85% of the households rely on firewood for cooking, and a significant amount of firewood comes from natural forest resources [27].

Environmental degradation in the region is primarily linked to recent development projects, namely significant infrastructure developments and investment projects [28]. Two of the most prominent projects are the crude oil offloading terminal and its onshore oil pipelines, operated by the China National Petroleum Corporation (CNPC) on Maday Island and an onshore gas terminal linked to an onshore gas pipeline operated by Posco Daewoo Myanmar Limited under the umbrella of Shwe Consortium on Ramree Island [15]. The onshore gas terminal receives natural gas from the Shwe production platform, located offshore on the edge of the continental shelf. The 40-inch diameter onshore gas pipeline originates at the first gas receiving station. A crude oil terminal is located on Maday Island with 12 crude oil storage tanks. The 771 km long pipeline links Maday Island to China's Yunnan province, transporting 22 million tons of oil annually [29]. The crude oil pipeline is laid in parallel with the gas pipeline. Both pipelines transverse through Rakhine state, Magwe division, Mandalay Division, and Shan State before entering China [8]. Fig 2 shows very high-resolution satellite images of the pipelines during pre- and post-construction periods.

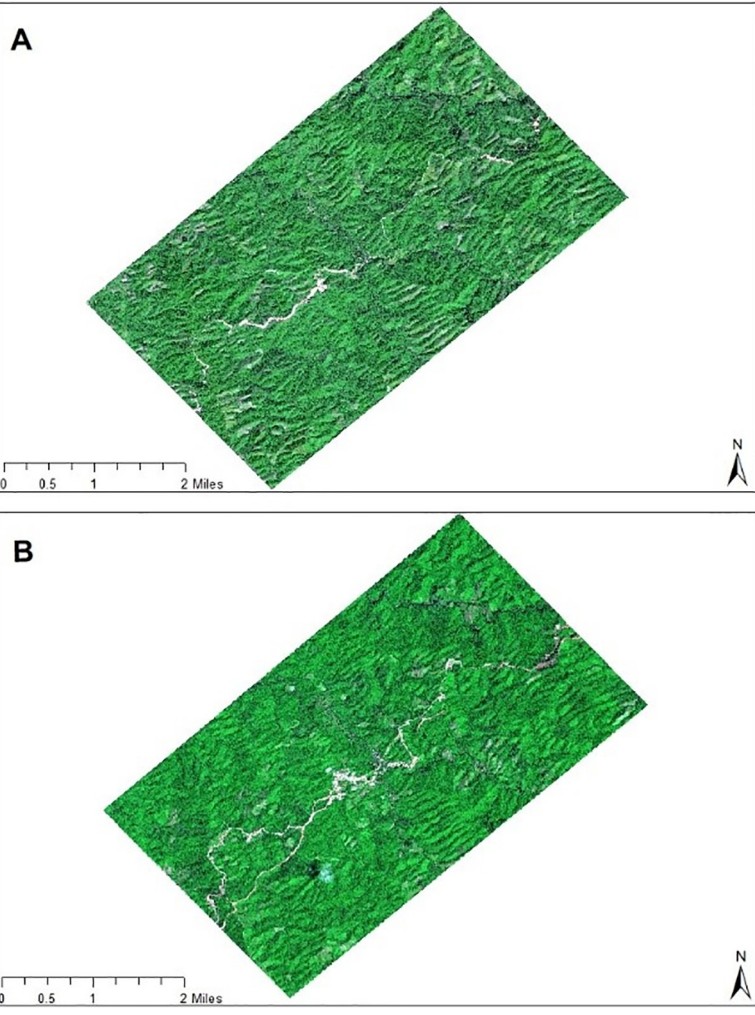

**Fig 2. True color composite map of the study area pre- and post-construction of the China-Myanmar Oil and Gas pipelines.** Image A is from 17[th] November 2010, showing the area prior to the construction of the pipelines. Image B is from 17[th] October 2012, showing the area after the construction of the pipelines. (DigitalGlobe order number: 059416456010_01) Source: http://www.digitalglobe.com.

The facilities were constructed in 2010 and have been in operation since 2013. However, the government and the project proponents have yet to solve widespread local tension featuring regular protests and widespread opposition stemming from environmental damage, land confiscation, and land compensation practices for ROW [30]. An initial EIA of the pipelines identified several potentially adverse environmental impacts in connection with the total clearance of scrublands, grassland, degraded and secondary woodland, and primary forests [27] One of the most pressing environmental issues is on land resources and livelihood. Loss of ecosystem services provided by the forests and woodlands and the loss of agricultural land are deep-seated concerns among local communities.

Geographically, the area examined in this paper extends from 19° 49'41.63" N (elevation 64m) to 19° 52'26.77" N (elevation 198m) latitude and 94° 03'20.57" E (elevation 333m) to 94° 06'42.03" E (elevation 133m) longitude. The measurement of maximum extension is 7.49 km in the west-east direction and 4.49 kilometers in the north-south direction. The total affected area is 35.39 km$^2$. The study area covers pipelines and their ROW, which is 30m in width for these particular pipelines [31].This area is densely forested with diverse land cover types, and the pipelines can be seen to extend across the area. Although a relatively small study area is chosen for analysis due to budget constraints, it represents the different land cover types of the Rakhine State. The study area was chosen based on the visibility of pipelines (as most parts are covered by concrete), the availability of the high-resolution satellite data and environmental vulnerability. Furthermore, the area is in proximity to human settlements, representing an area with active local environmental groups opposed to the pipelines. The environmental, social, and economic conditions of the study area are also representative of other areas along the pipelines in Rakhine State.

## Data and image processing

Diverse datasets are used, including geospatial, socio-economic, demographic, and biophysical data to represent land use and forest cover changes to assess the impact of pipelines on both, ecosystems and livelihoods. The main satellite data used for the classification of Land-Use-Land-Cover-Change (LULCC) are commercial VHRI orthorectified multispectral satellite images, GeoEye-1 and Worldview-2. The satellites are two of the world's highest resolution commercial earth-imaging satellites and offer imageries with 0.5m resolution. GeoEye-1 satellite's positional accuracy is the best of all available satellites today [32]. VHRI have been widely used in land use classification, environmental monitoring and urban planning [33]. Previous studies quantifying land cover changes in Myanmar mostly used freely available satellite images such as Landsat satellite imageries to produce countrywide forest maps [34] [24]. VHRI Pléiades satellite images (70-cm panchromatic and 2.8-m multispectral) are utilized to identify changes in land use categories in the Tanintharyi Region, in south-eastern Myanmar [35]. When using low and medium resolution data, some studies combined multisensors, such as optical, radar and hyperspectral satellite data to improve classification accuracy [36–38]. The use of VHRI is deemed sufficient to easily distinguish between different land cover classes accurately.

Spatial data obtained from the CNPC were used as reference data to manually identify the exact location and route of the pipelines. These were digitized, using Google Earth. In addition, we collected images from Landsat 5, 7, and 8 for 2005, 2010, and 2012 for four representative scenarios in three districts (Kyaukphyu, Mandalay, and Minbu) to validate the pipeline route and the year of construction (see Fig 3). We collected and stacked a total of best available 12 Landsat scenes (cloud cover <10%, acquisition between November-February), covering four tiles (Paths: 132–134 and Rows: 044–047), using Landsat bands that record surface reflectance

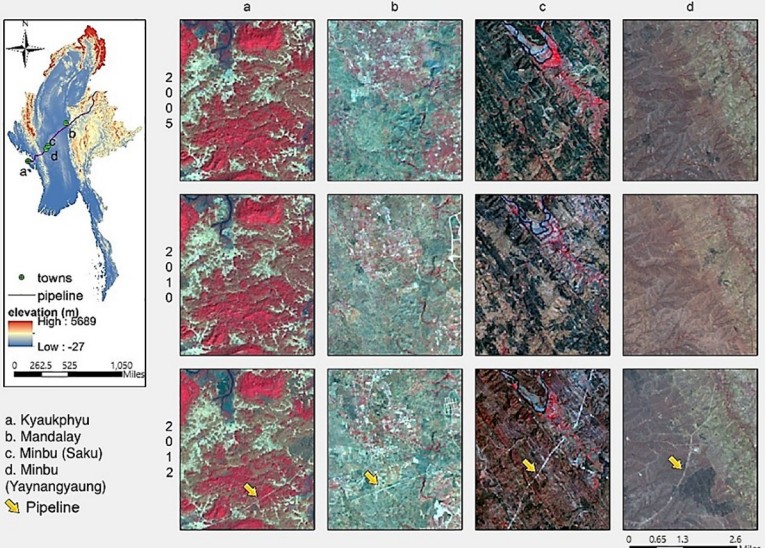

**Fig 3. False-color composite images of the three sites between pre- and post- pipelines construction.** The yellow arrow shows the appearance of the pipelines in 2012 images traversing all three locations. 2005: Landsat 5(R:3, G:3, B:1), 2010: Landsat 7(R:3, G:3, B:1), 2012: Landsat 8 (R:4, G:3, B:2). (Landsat-5, 7and 8 images courtesy of the U.S. Geological Survey; https://earthexplorer.usgs.gov/).

in the visible, [34] near-and mid-infrared spectrum and that have a minimum 30-m resolution. They are a subset to the study area's geographic boundaries. Radiometric calibration and FLAASH (Fast Line-of-sight Atmospheric Analysis of Spectral Hypercubes) atmospheric correction were applied to the original Landsat images to obtain the ground surface reflectance (ρ) in ENVI. Experimental districts for land cover classification were selected, based on the background information on the severity of the impacts, the proximity of pipelines to human settlements, and the possibility of field data collection. Most of the areas along the pipelines are logistically challenging and potentially dangerous to visit.

Concurrent with the digitization, we conducted field visits to selected sites, during February 2019, for verification and the collection of training samples (Fig 4). Field trips were permitted by the Minister of Electricity, Industry and Transportation based in Sittwe, Rakhine state. Field visits allow for reliable site observation and real-time documentation of the conditions of land cover and land use, pipeline area boundaries, and the surrounding environment. This spatial data was then compared with previously digitized data in Google Earth. Training sites for land cover classification were determined and experimental satellite datasets were derived. Land cover classification of the study area was determined, based on the existing Myanmar land-use maps developed by the United Nations Environmental Program (UNEP). Besides, countrywide forest cover change data produced by [34] was useful for cross-checking forest land classification in the region. Google Earth is widely used for collecting training and validation data for remotely sensed projects, especially when field data collection is difficult [34]. In our case, we combined ground data with randomly chosen samples and manually digitized 370 training polygons distributed throughout the study area to cover the entire satellite image. The training sample covers a total of five land cover and land use categories; (1) forest, (2) scrubland, (3) infrastructure development, (4) residential areas, and (5) agricultural land. This method assures the representation of the samples for each land category [39]. Using a large number of reference data can enhance the most accurate classification outcome in nonparametric machine learning classification, such as Random Forest [40]. Training samples and

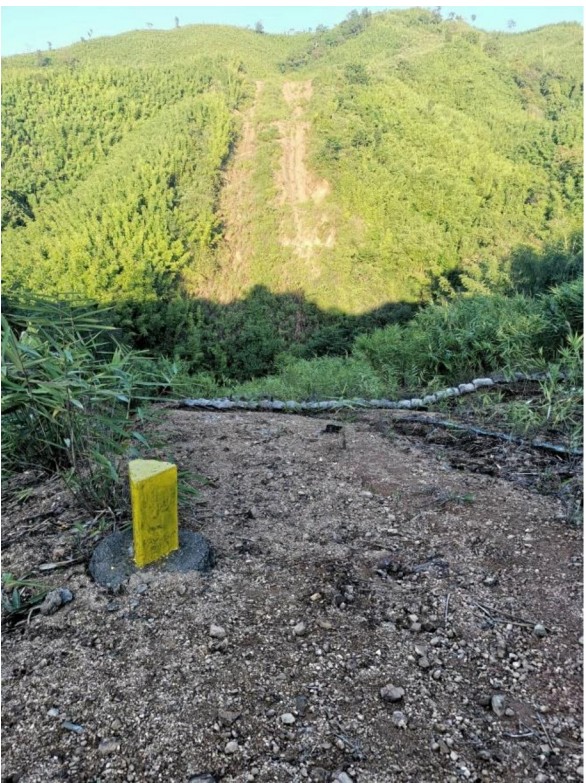
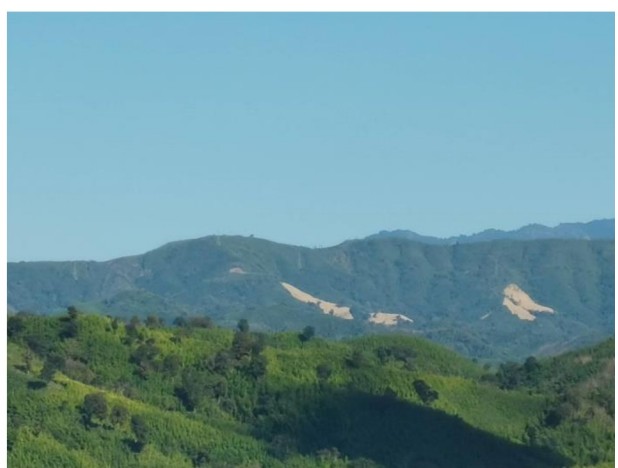

**Fig 4. Examples of field visits used to help guide the development of training polygons and validation.** (Source: Authors' own).

reference data were later used as input variables for the calibration in the Random Forest model. Following the method used by [41], we assigned a random number between 0 and 1. Polygons with assigned values of ≤0.70 were assigned as training data and polygons with >0.70 as testing data. Random numbers between 0 to 1 were then assigned to each pixel. The testing polygons were equally divided for testing and validation. This resulted in 255 training polygons and 115 testing polygons.

To quantify Land-Use-Land-Cover-Change (LULCC) associated with the pipelines' operation, primary data from the existing time series from the two commercial VHRI orthorectified satellite images, GeoEye-1 (GeoProfessional) and Worldview-2 were collected [42]. Data captured very close dates for two years in the same study area. To compare pre-operation, operation, and post-operation periods, images from two acquisition dates (t0 = 2010 November 17, t-1 = 2012 October 17) with a spatial resolution of 0.5m (multispectral bands) were used for our main study area in Ann township. These images were mainly used for tracking and classifying land cover changes in the area of immediate proximity to the pipelines, using the RF classification method. Cloud free GeoEye-1 image was available for the year 2010, and Worldview-2 was suitable for 2012. GeoEye-1 image is multispectral with four bands (blue: 450–510 nm, green: 510–580 nm, red: 655–690 nm, near-IR: 780–920 nm), acquired on 17 November 2010. The worldview-2 image also contains multispectral bands (4 standard colors: red: 624–694 nm, blue: 442–515 nm, green: 506–586 nm, near-IR: 765–901 nm), acquired on 17 October 2012. The (pre- and post-operation) dates were chosen on the basis of cloud-free images for the study areas. Due to the monsoon climate patterns throughout Myanmar, it is challenging to obtain cloud-free satellite imagery. Image fusion was performed using the Gram-Schmidt Pan Sharpening method. No additional orthorectification was required for both images [43].

Next, a supervised classification, using Random Forest (RF) in R statistical software's the "randomForest" package was performed [44]. Very high-resolution remote sensing imagery and advanced image classification algorithms allow data mining and a precise assessment of a range of target features on the ground [45]. RF classification is a non-parametric machine learning algorithm widely used in remote sensing and classification modeling throughout different disciplines and objectives [46–48]. RF is a robust model particularly suitable for land cover and land use classification as it can effectively process a large number of predictor variables as well as complex datasets [49, 50]. The RF model was found to outperform other traditional parametric based image analyses due to its capacity to deal with missing values and complex variables, and high overall classification accuracy [51, 45, 52]. Another significant advantage of the RF is that it creates an ensemble of trees, each providing a "vote" to select the best classification approach. That is, the majority of votes from the assemblages of the tree created in Random Forest decide the class assignment of the pixel, and the results of a large number of trees are aggregated internally [53]. As RF models require specification of several parameters to execute the model, each RF tree was built by training each tree in the forest (ntree) with the number of input predictor-variables (mtry), randomly chosen at each split from the training dataset [54]. The number of predictor-variables was set at the square root of input variable (i.e. 4 bands), therefore the number of variables tried at each split was 2. Following the recommendations from previous studies, a large number of trees (n = 1000) was selected to run the RF algorithm to stabilize the mean squared error in each iteration process [55]. Given the small number of variables used in our current study, we calibrated the model based on the complete dataset and produced land cover maps with five pre-defined land cover classes for the two time periods. At the last step of the classification process, land cover maps were converted into polygon shapefiles in ArcGIS for further analysis.

Finally, land cover land-use change was calculated, and maps were generated, using the Land Change Modeler (LCM) in TerrSet. LCM is the land planning and decision support system that simplifies the complexities of change analysis and allows for rapid analysis of land cover change and model relationships between variables of interest. LCM is an established methodology widely applied in spatially explicit LULCC modeling, trend change analysis, and scenario analysis [56–60]. LULCC detection in LCM has proven to be more accurate than other modeling tools [56, 57]. The LCM was used to model land cover and land use detection along the pipelines within the study area, based on the spatial patterns from 2010 to 2012; especially land cover transitions from forest to agricultural land, agricultural land to forests, infrastructure development to forests, residential area to forests, scrubland to forests, forest to infrastructure development, forests to residential area and forests to scrubland. The model also calculates gains and losses, as well as project net changes, and determines drivers of change for each land cover category, both, in map and graphical form.

A wide range of geospatial information, such as the pipelines' geographical location, road network, other physical features, affected villages and the geographic boundaries of villages and townships were derived from the databanks of The Humanitarian Data Exchange and Myanmar Information Management Unit's GIS resources.

For the accuracy assessment of the model, RF classification internally estimates accuracy during the bootstrapping process [51]. Accuracy assessment quantifies the accuracy of maps, estimates the area of each class defined by reference classification, and assesses uncertainty of area classifications [61]. The accuracy was assessed, based on the validation score approach to validate the RF model. The validation score is calculated by setting a part of the original training data aside before training the models and using the decision trees of the ensemble [62]. Classification accuracy was expressed by reporting the estimated confusion matrix in terms of overall accuracy, commission error (user's accuracy), and omission error (producer's

**Table 1. Accuracies based on an accuracy assessment of the two LULCC maps for 2010.** Rows are map categories, and columns are reference categories.

| Category | Year | Agriculture | Forest | Infrastructure Development | Residential Area | Scrubland | Total | Commission |
|---|---|---|---|---|---|---|---|---|
| Agriculture | 2010 | 28 | 506 | 16 | 5 | 70 | 625 | 1.8 |
| Forest | 2010 | 130 | 6091 | 90 | 40 | 461 | 6812 | 25.8 |
| Infrastructure | 2010 | 13 | 381 | 124 | 11 | 41 | 570 | 1.2 |
| Residential Area | 2010 | 7 | 236 | 12 | 11 | 25 | 290 | 0.6 |
| Scrubland | 2010 | 52 | 1429 | 27 | 13 | 182 | 1730 | 6.2 |
| Total | 2010 | 229 | 8643 | 269 | 80 | 779 | 10000 | 35.7 |
| Omission | 2010 | 6.0 | 7.2 | 4.8 | 2.7 | 15.0 | 35.7 | 0 |

accuracy). The column of the matrix is the reference information, the row is the information of the classification result, and the intersect gives the number of samples classified into a specific class [63]. Overall accuracy refers to the proportion of samples that are correctly classified and user's accuracy indicates the proportion of samples measured as each class.

In addition, we followed the method developed by [61] for assessing land cover accuracy. As recommended by [61], we adopted a stratified random sampling design. The required sample size was calculated, using the following formula:

$$n = \frac{\left(\sum W_i S_i\right)^2}{[S(\hat{O})]^2 + \left(\frac{1}{N}\right)\sum W_i S_i^2} \approx \left(\frac{\sum W_i S_i}{S(\hat{O})}\right)^2$$

Where n = number of units, $S(\hat{O})$ is the standard error of the estimated overall accuracy, Wi is the mapped proportion of the area of class $i$ and $S_i$ is the standard deviation of $i$. $S_i = \sqrt{U_i(1 - U_i)}$. We specify a target standard error for overall accuracy of 0.01. Using proportional approach, we allocated sample size of 10–50 for each change strata. A small overall testing sample size allows for only 10 sample units for some stratum. The estimated variances are them computed based on the sample size allocation.

## Results

### Accuracy assessment

The aim of an accuracy assessment was to evaluate the ability of a model for detecting and delineating LULCC within a study area. Tables 1 and 2 summarize the classification accuracy validation results of the LULCC maps obtained from the RF model. The overall accuracy of the classification was 64.33% for 2010 and 65.28% for 2012. The Confidence Interval (CI) for both years is 0.95. This relatively low overall accuracy can be due to the use of a small study area and/or small training samples. Previous research has suggested that high spatial

**Table 2. Accuracies based on an accuracy assessment of the two LULCC maps for 2012.** Rows are map categories, and columns are reference categories.

| Category | Year | Agriculture | Forest | Infrastructure Development | Residential Area | Scrubland | Total | Commission |
|---|---|---|---|---|---|---|---|---|
| Agriculture | 2012 | 32 | 534 | 13 | 11 | 50 | 640 | 1.9 |
| Forest | 2012 | 176 | 6016 | 109 | 42 | 428 | 6771 | 25.1 |
| Infrastructure | 2012 | 11 | 313 | 241 | 12 | 17 | 594 | 1.6 |
| Residential Area | 2012 | 7 | 216 | 22 | 15 | 26 | 286 | 1.0 |
| Scrubland | 2012 | 39 | 1418 | 12 | 13 | 164 | 1646 | 5.2 |
| Total | 2012 | 265 | 6497 | 397 | 93 | 685 | 9937 | 34.7 |
| Omission | 2012 | 5.9 | 7.4 | 3.6 | 2.6 | 15.2 | 34.7 | 0 |

**Table 3. Error matrix (sample count) for 2010 based on stratified random sampling.**

|  | Reference | | | | | Total | Area | $W_i$ |
|---|---|---|---|---|---|---|---|---|
|  | Agriculture | Forest | Infrastructure | Residential | Scrubland | | | |
| Agriculture | 7 | 4 | 2 | 1 | 1 | 15 | 74 | 0.062 |
| Forest | 9 | 32 | 5 | 3 | 1 | 50 | 690 | 0.581 |
| Infrastructure | 5 | 2 | 8 | 0 | 0 | 15 | 69 | 0.058 |
| Residential | 2 | 1 | 0 | 6 | 1 | 10 | 41 | 0.035 |
| Scrubland | 4 | 1 | 1 | 3 | 16 | 25 | 315 | 0.265 |
| Total | 27 | 40 | 16 | 13 | 19 | 115 | 1,189 | 1 |

heterogeneity and small sample sizes will result in much lower classification accuracies [64]. The residential area is the most accurately classified category for both years, with 99.4% and 99% for 2010 and 2012, respectively. Infrastructure development also had very high accuracy rates, with 98.8% in 2010 and 98.4% in 2012. The third highest accuracy rate was associated with agricultural land, 98.2% for 2010, and 98.1% for 2012. Forests had the lowest accuracy rate; 74.2% for 2010, and 74.9% for 2012; followed by scrublands, which had an accuracy rate of 93.8% for 2010 and 94.8% for 2012. Forests might sometimes be assigned falsely as scrublands and vise vasa during the training digitization process due to their visual similarities, leading to either over- or underestimation. Conversely, residential areas, and infrastructure development areas can be comparatively clear and can be accurately identified.

The error matrices obtained from stratified random sampling design are presented below. The results contain stratified estimation of area for each class. Tables 3 and 4 displays error matrices in sample count and Tables 5 and 6 estimate the area proportion. Based on the results from stratified random sampling method, the overall accuracy of the classification was 62.16% for 2010 and 61.86% for 2012. The tables also display adjusted area estimate in hectares.

## Land use and land cover change along the pipelines

Maps of the study area from two points in time were analyzed, including pre- (2010) and post-construction (2012) periods. This way LULCC along the China-Myanmar Oil and Gas Pipelines were established. The maps include designations of five major land cover classes; forests, agriculture, infrastructure development, residential /non-forest areas, and scrubland. The resulting maps for the land-cover classification of pre- and post-construction periods, using

**Table 4. Error matrix (area proportion) for 2010 based on stratified random sampling.**

|  | Reference | | | | | Total | Area | $W_i$ |
|---|---|---|---|---|---|---|---|---|
|  | Agriculture | Forest | Infrastructure | Residential | Scrubland | | | |
| Agriculture | 0.0289 | 0.0165 | 0.0082 | 0.0041 | 0.0041 | 0.0619 | 74 | 0.062 |
| Forest | 0.1045 | 0.3715 | 0.0581 | 0.0348 | 0.0116 | 0.5805 | 690 | 0.581 |
| Infrastructure | 0.0194 | 0.0077 | 0.0310 | 0.0000 | 0.0000 | 0.0581 | 69 | 0.058 |
| Residential | 0.0070 | 0.0035 | 0.0000 | 0.0209 | 0.0035 | 0.0349 | 41 | 0.035 |
| Scrubland | 0.0423 | 0.0106 | 0.0106 | 0.0318 | 0.1694 | 0.2647 | 315 | 0.265 |
| Total | 0.2020 | 0.4099 | 0.1079 | 0.0916 | 0.1886 | 1.0000 | 874 | 0.73535 |
| Area [ha] | 177 | 358 | 94 | 80 | 165 | 874 | | |
| Standard Error | 0.0488 | 0.0455 | 0.0318 | 0.0390 | 0.0453 | | | |
| User's | 0.47 | 0.64 | 0.53 | 0.60 | 0.00 | | | |
| Producer's | 0.14 | 0.91 | 0.29 | 0.23 | 0.02 | | | |
| Overall | 0.6217 | | | | | | | |

**Table 5. Error matrix (sample count) for 2012 based on stratified random sampling.**

| | Reference | | | | | Total | Area | W$_i$ |
|---|---|---|---|---|---|---|---|---|
| | Agriculture | Forest | Infrastructure | Residential | Scrubland | | | |
| Agriculture | 7 | 4 | 2 | 1 | 1 | 15 | 75 | 0.063 |
| Forest | 9 | 32 | 5 | 3 | 1 | 50 | 673 | 0.566 |
| Infrastructure | 5 | 2 | 8 | 0 | 0 | 15 | 96 | 0.081 |
| Residential | 2 | 1 | 0 | 6 | 1 | 10 | 54 | 0.045 |
| Scrubland | 4 | 1 | 1 | 3 | 16 | 25 | 291 | 0.245 |
| Total | 27 | 40 | 16 | 13 | 19 | 115 | 1,189 | 1 |

the preprocessed satellite images are shown in Fig 5. Based on the two maps, net change in the area of each LULC category was calculated. Results are shown in Table 7. Overall, on both maps, forests are the most extensive land cover in the area, followed by scrubland and agricultural land. Forests cover an area of 690 hectares in 2010, i.e. over 60% of total land cover. Time series land cover maps and statistics (Fig 5 and Table 3) reveal that there is a notable increase in infrastructure development over the two years. This can be associated with the construction of pipelines and the associated facilities in the area. As can be seen in Fig 5, the pipelines diagonally crossing the study area are the most solid form of infrastructure development. The only infrastructure development seen in the 2010 map (A) is the Minbu-Ann highway passing through the area. As a result of the pipeline construction, the total area of infrastructure development expanded very rapidly across the region, from 59 hectares to 86 hectares, i.e. the total growth rate is 44.65%. The forest cover within the 2 km area along the pipelines in the study area shows a downward trend, from 690 hectares in 2010 to 673 hectares in 2012 with a total forest loss of 17 hectares and a net decline of -2.45% just in two years.

Forest loss overall is driven primarily by pipeline construction and a slight increase in residential areas—from 54 hectares to 59 hectares (1% net increase). The increase in residential area is mostly triggered by pipeline activities and most of the change is associated with construction and pipeline maintenance workers [65]. Overall, non-forest related activities have witnessed a net gain of 13 hectares around the pipelines. Most of the agricultural land remains unchanged, with only 1.5 hectares of additional agricultural land being created. Scrubland areas make up 20% of the area's total land cover and are declining more rapidly than other land cover classes and there has been a net loss of 24 hectares (10.74% decline rate).

**Table 6. Error matrix (area proportion) for 2012 based on stratified random sampling.**

| | Reference | | | | | Total | Area | W$_i$ |
|---|---|---|---|---|---|---|---|---|
| | Agriculture | Forest | Infrastructure | Residential | Scrubland | | | |
| Agriculture | 0.0294 | 0.0168 | 0.0084 | 0.0042 | 0.0042 | 0.0631 | 75 | 0.063 |
| Forest | 0.1019 | 0.3623 | 0.0566 | 0.0340 | 0.0113 | 0.5660 | 673 | 0.566 |
| Infrastructure | 0.0269 | 0.0108 | 0.0431 | 0.0000 | 0.0000 | 0.0807 | 96 | 0.081 |
| Residential | 0.0091 | 0.0045 | 0.0000 | 0.0272 | 0.0045 | 0.0454 | 54 | 0.045 |
| Scrubland | 0.0392 | 0.0098 | 0.0098 | 0.0294 | 0.1566 | 0.2447 | 291 | 0.245 |
| Total | 0.2065 | 0.4042 | 0.1179 | 0.0948 | 0.1767 | 1.0000 | 1,189 | 1 |
| **Area [ha]** | 246 | 481 | 140 | 113 | 210 | 1,189 | | |
| Standard Error | 0.0486 | 0.0454 | 0.0315 | 0.0398 | 0.0432 | | | |
| **User's** | **0.47** | **0.64** | **0.53** | **0.60** | **0.00** | | | |
| **Producer's** | **0.14** | **0.90** | **0.37** | **0.29** | **0.03** | | | |
| **Overall** | **0.6186** | | | | | | | |

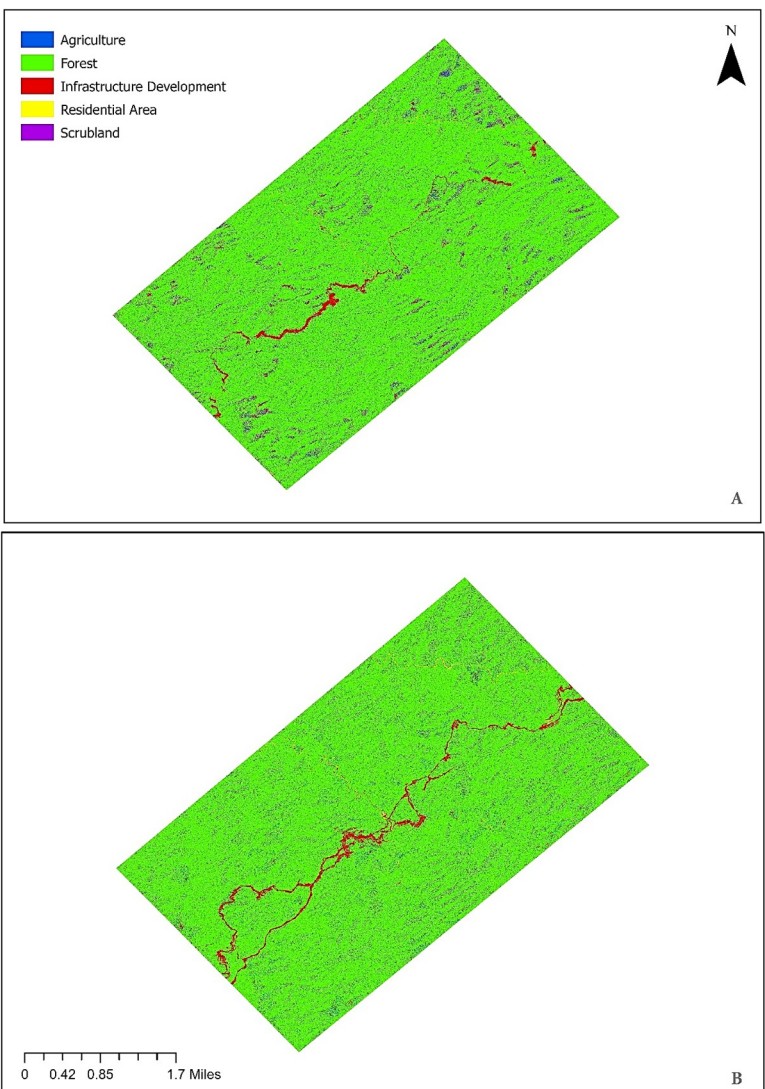

**Fig 5.** (A)2010 Pre-construction and (B)2012 Post-construction land cover map for the study area classified into five major land cover classes, including agriculture (orange), forest (green), infrastructure development (red), residential arear/non-forest (pink), scrubland (blue).

Development of scrublands is likely to have been affected by operation of the pipeline as the destructive form of land use associated with construction activities can significantly alter vegetation.

**Table 7. Area (in hectares) and spatial change in land cover land use classes and overall net gain and losses between 2010 and 2012 in the study area.**

| Class Name | Acquisition Date | | | | Net LULC Change | Growth/Decline Rate |
|---|---|---|---|---|---|---|
| | 2010 (ha) | | 2012 (ha) | | | |
| | Area | Percent | Area | Percent | | |
| Agriculture | 73.5288 | 6% | 75.0119 | 6% | 1.4831 | 0 |
| Forest | 690.0288 | 65% | 673.1166 | 63% | -16.9122 | -2.45% |
| Infrastructure development | 69.3524 | 6% | 95.8544 | 8% | 26.502 | 44.65% |
| Residential Area | 41.449 | 4% | 54.1524 | 5% | 12.7034 | 1% |
| Scrubland | 314.5627 | 20% | 290.7004 | 18% | -23.8623 | -10.74% |

**Table 8. Gain and losses, and net change of land cover between 2010 and 2012.**

| Class | Gains and Losses | | Net Change |
|---|---|---|---|
| Agriculture | -68 | 69 | 1 |
| Forest | -376 | 359 | -17 |
| Infrastructure development | -51 | 77 | 27 |
| Residential area | -23 | 36 | 13 |
| Scrubland | -253 | 229 | -24 |

## LULCC modelling results

The increase and decrease of each land use and land cover class can be established more accurately with land change analysis results obtained through the Land Change Modeler (LCM) in TerrSet. Table 8 shows gains and losses for each land use in 2010 and 2012. Quantification of land use and land cover changes surrounding the pipelines from the classification in the RF model is supported by the results generated by LCM in TerrSet. With regards to agriculture use, whilst there is only a 1% net change, more than 90% of agricultural lands have experienced changes, with a loss of 68 hectares on the one hand and a gain of 69 hectares on the other. This change is explained by extensive agricultural land-use transitions driven by the construction of the pipelines, as well as associated corridors, and other facilities. Much agricultural lands in the region was confiscated, and other land cover types such as forests and scrublands were converted into agricultural land during the study period to make way for the pipelines and ROW [3]. Pipelines construction-related activities also caused damage to and destruction of farmland. As a result, other land cover types were shifted into farmland to compensate farmers. Therefore, the net change in forests and scrublands is significant, at 60–80% conversion rate. The amount of area converted from forest and scrubland into other land use types is 376 and 253 hectares, respectively.

Other transitions in land use reflect infrastructure development activities with the amount of the area transferred to infrastructure development projects being 77 hectares. The amount of infrastructure development area transferred to other land-use types is 51 hectares. This result can be explained by many pre-existing human activities such as village and mountain roads being moved due to pipeline operation related land use [14]. Moreover, the residential area in the region also experienced a dramatic transformation during the study period. More than 56% (23 hectares) of residential area has been transferred into other land-use types. This change is due to the large-scale confiscation of residential lands for the pipelines and the replacement of other land covers for residential areas [66]. Overall, the result shows an accumulated increase in the land area of agriculture, infrastructure development, and residential area. However, there is a declining trend in forests and scrublands over the two study periods. Forest and scrubland areas witnessed the maximum transformation into human activity, whereas residential areas contributed to the smallest extent.

The LULCC map with major land conversion classes, depicting land cover conversion and non-conversion between 2010 and 2012 along the pipelines is shown in Fig 6. The map identifies a few specific characteristics of LULCC for each land cover and land use type. Table 9 summarizes the occurrence of forest transformation in the study area. Change detection analysis identifies changes of particular magnitude by excluding transition of less than 10 hectares. As a result, all land cover and land use transition reported only included forest conversion. This result suggests that all major transitions occurring in the region were associated with forest areas. The land cover conversion also indicates that forest to other land cover types conversion is the most extensive type of transformation (see Table 5). 678 hectares of forest

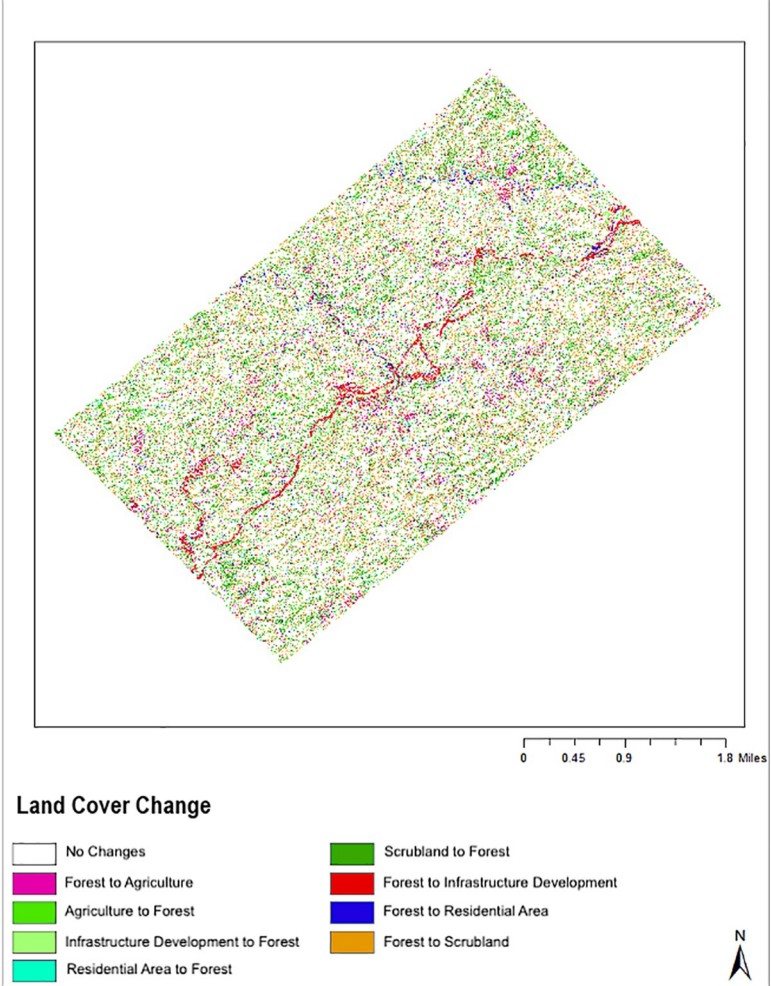

**Fig 6. LULCC map with major land conversion classes, depicting land cover conversion and non-conversion between 2010 and 2012 along the pipelines in Ann township.**

were transformed for the infrastructure development, 60.5 hectares into agriculture, 31 hectares intro residential area, and 219 into scrubland between 2010 and 2012.

Conversely, the transformation of scrubland was mostly associated with forests, accounting to 238 hectares. Additionally, 59 ha of agricultural land, 44.5 ha of infrastructure development

**Table 9. LULC conversion between 2010 to 2012.**

| Category | Hectares | Legend |
| --- | --- | --- |
| 1 | 60.479857 | Forest to Agriculture |
| 2 | 59.210479 | Agriculture to Forest |
| 3 | 44.440748 | Infrastructure development to Forest |
| 4 | 19.487422 | Residential area to Forest |
| 5 | 237.887966 | Scrubland to Forest |
| 6 | 67.835980 | Forest to infrastructure development |
| 7 | 31.210960 | Forest to Residential area |
| 8 | 218.899505 | Forest to Scrubland |

areas, and 19.5 ha of residential areas were converted into forests. This can be attributed to the recent reforestation efforts by the forest department in order to achieve the country's goal of restoring 12 million hectares of forests by 2030 [67]. While a large area of forest is also transforming into other land types, a significant amount of scrubland and agricultural land is converted into forest land. Scrubland and farmland conversion are likely to be the result of the government's initiatives to transform scrubland back into forest land and to reduce forest degradation due to human activities. However, it is important to note that the conversion of residential areas and other anthropogenic activities into forests is still small compared to forest land being converted into other land-use types.

Fig 7 shows forest gains and losses below 10 hectares. Transformational gains and losses of forests were found to be highly interlinked with infrastructure development. Most of the forest losses occurred in small patches with less than 10 hectares. The further away forested areas are from the infrastructure development, the smaller the change. Thus, a large number of forest areas remained unchanged in areas far away from the pipelines. Hence, the highest overall losses / large-scale forest losses with more than10 hectares occurred in areas closer to the infrastructure development, owing to forest transition to pipeline related construction and to exposure to very high environmental pressure from surrounding activities. Generally speaking, a substantial area of forested land is lost annually due to human activities and economic development [68].

Accordingly, substantial forest conversions took place mainly within the center of the study area where concentrated infrastructure development activities are located, indicating a direct link between the location of the pipelines and the large-scale forest decline. In the center of the study area, surrounding the pipelines, forests to infrastructure development conversion spread more southward, and forest to residential area conversion occurred in the northern part of the study area. Forest to agricultural land and forest to scrubland alterations are concentrated in the center of the map. Meanwhile, the diagrams for overall forest change to all other LULC types and other land types to forests show similar trends with the majority of change happening in the immediate proximity to the pipelines. In the outermost parts of the study area, significantly fewer LULC changes have occurred compared with the center part where the pipelines are located. These results confirm that the magnitude of the impact of infrastructure development by the pipeline are apparent within a 2 km distance / radius. Although the intensity of change can be different given the diverse ecosystem conditions and land cover types across Myanmar, similar occurrence of LULCC can be predicted in other areas along the pipelines. However, future scenarios can change depending on the planning and implementation of mitigation measures to protect forests.

## Discussion

Very-high-resolution satellite data were used in the RF classification method and land change modeler to derive detailed LULCC information for analyzing deforestation and afforestation conversion patterns along the China-Myanmar Oil and Gas pipelines between pre-construction (2010) and post-construction (2012) periods. Over the two years, the five investigated land-use types underwent substantial changes along the pipelines. Notably, forests experienced a rapid decline and several conversion patterns. From 2010 to 2012, a large area of forests was converted into anthropogenic use, agricultural land, and scrubland (deforestation) and an extensive area of scrubland and agricultural lands was converted into forests (due to afforestation efforts). The deforestation process mostly resulted from the expansion of infrastructure development (i.e., the construction of pipelines and related activities). Previous studies on the impact of pipelines on forests also found that pipelines contributed to forest losses, although the extent of the impact is likely to depend on the route and the width of the ROW ([69].

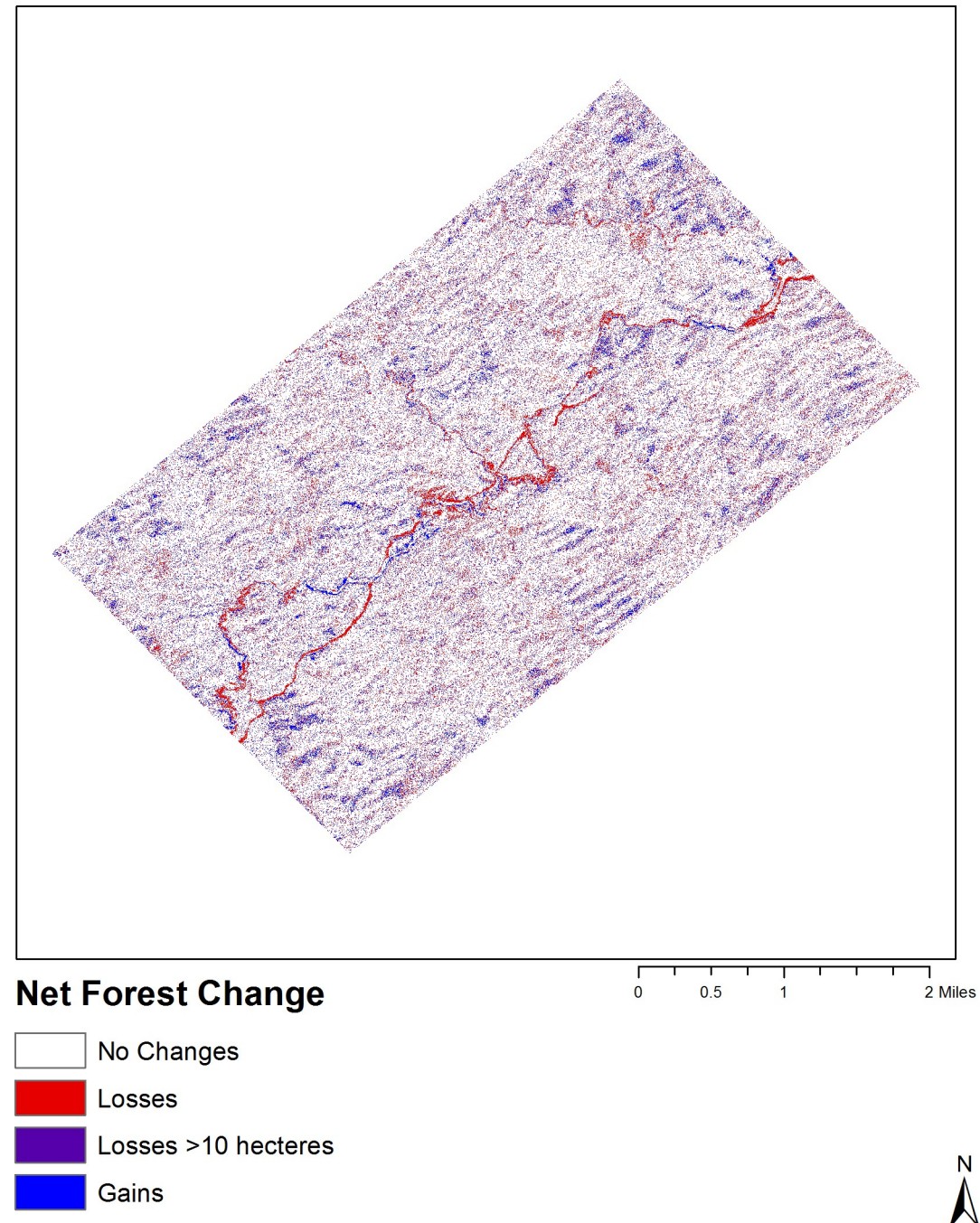

**Net Forest Change**

☐ No Changes

🟥 Losses

🟪 Losses >10 hecteres

🟦 Gains

0    0.5    1    2 Miles

N

**Fig 7. Net forest change between 2010 and 2012.**

Anthropogenic drivers, such as the construction of large-scale pipeline projects, are bound to create significant LULC transitions for all land cover classes [57]. In Myanmar, infrastructure development, agricultural expansion, fuelwood production, and illegal logging are the main drivers of forest cover loss and degradation [70]. Built-up areas are also reported as a major threat for mangrove deforestation in Myanmar [41]. However, considering the intensity

of forest losses occurring within the immediate proximity of the pipelines and their ROW, all forest losses can be attributed to the pipelines' construction.

Afforestation at the expense of scrublands and croplands is connected with several forest protection regulations. During the past few years, in a bid to restore forested areas, several regulations were launched in Myanmar, including the National Forest Master Plan, National REDD+ Strategy, and the Myanmar Reforestation and Rehabilitation Program (MRRP). These programs promoted large-scale plantations over the last decade throughout the country, with an estimated 567,000 ha of private commercial plantations. Plantations offer attractive investment opportunities and contribute to meeting the country's reforestation targets. However, it is important that most of these forests constitute lower-value fast-growing wood such as eucalyptus, acacias, fuelwood, and pulp [70]. Although fast-growing plantations can provide a substantial amount of timber within a short time, they are frequently associated with negative environmental and social consequences, such as replacing natural forests, decreasing water availability, depletion of biodiversity, and encroachment on agricultural lands [71]. Eucalyptus plantations in particular can cause various environmental issues such as desertification, biodiversity loss, and water deprivation due to its rapidly growing nature and high fertilizer consumption [72]. Large-scale development of fast-growing trees tends to aggravate logging and further increases the conversion of intact forests into commercial plantations. In Borneo, for example, annual commercial plantation expansion has been found to be positively correlated with annual forest loss [73]. It is also evident that land use in some regions of Myanmar has shifted to rubber, betel nut, cashew and oil palm [35], [74], all of which needs conversion of abundant land areas [75].

As shown in Fig 7, areas closer to infrastructure development tended to experience higher rates of transitions from forests, scrublands and agricultural land to other land use types, while further away areas were less likely to convert from tree-covered into built-up areas. Apart from the pipelines, the expansion of residential areas also had a strong influence on all LULC change classes. Not surprisingly, the areas near the pipelines experienced some of the sharpest forest reductions, with more than 10 hectares of forest loss throughout the study area. Forest decline accelerates with increasing development of infrastructure projects in densely forested areas where forests are often severed and fragmented to make way for projects. Although the experimental study area covered in this paper is small, the quantity and rate of LULC transition are significant according to the modelled LULCC. Looking at the anthropogenic development scenario, with a growth rate of 44.65%, forests and scrublands experienced a net decline of -2.45% and 10.74%, respectively, in just two years. Although deforestation and afforestation co-exist, the area of deforestation is still more extensive than afforestation. Generally speaking, forest cover loss in Myanmar has been accelerating over the years. There has been an overall annual decline of forests of 0.3% between 1990 and 2000 [24]. Between 2002 and 2014, annual forest loss increased to 0.55% [34]. It is likely that the impacts of oil and gas pipelines on LULC are similar along other pipelines in Myanmar, although different ecosystems and LULC types need to be considered. In this context, areas in other regions should also be investigated to achieve higher accuracy of modeling results. Incorporating local governments' policies and development plans into modeling processes might also increase accuracy for scenario modeling in other areas.

Although Myanmar's forests continue to decline, the country does not have appropriate forest management practices for forest restoration and sustainable agricultural use of [21]. Even though the government appears to have been using reforestation activities for several decades, these actions have not fully achieved desired results [76]. Also, forest regrowth does not necessarily bring back the original ecosystem which had been degraded [77]. Results presented in this paper suggest that portions of scrubland and agricultural lands are converted

into forests. This action will only bring short-term forest gain but not long-term sustainability. Forest restoration should focus on the reclamation of degraded and deforested areas, also to improve economic and environmental conditions of local communities. It is clear that at least some of the impacts of forest and land clearing for the pipelines can be predicted and minimized through better planning and management. However, more research is needed to better understand the impact of oil and gas exploration and associated infrastructure development on ecosystems, their services, and social and human rights.

## Conclusions

Over recent years, owing to unprecedented region wide economic development, Myanmar's land use and land cover have experienced substantial changes and dramatic forest loss [78]. Forest severance and fragmentation can be attributed to the construction of thousands of miles of oil and gas pipelines traversing the country's forested areas. Forest fragmentation occurs when large and continuous forested areas are broken into smaller patches of forest, typically due to human activities [79]. LULCC analysis provides vital information on environmental change, triggered by development projects, such as oil and gas pipelines. In this paper, detailed information of LULCC was provided, using very-high-resolution-satellite-imageries. This allowed for an analysis of forest and land cover conversion along the China-Myanmar Oil and Gas pipelines in a 35.39 $KM^2$ study area in Rakhine State of Myanmar from 2010 to 2012.

The paper addressed three critical questions: (1) What is the rate and pattern of LULCC along the pipelines? (2) How much forests have been lost during the study period? (3) What is the pattern of afforestation in the study area? Analysis reveals that forests have undergone continuous change and have witnessed a dramatic decline leading to the loss of 16.9 hectares of (-2.45% net decline) forest during this two-year period. LULCC included an expansion of anthropogenic disturbances in the form of pipelines construction and residential areas as well as a reduction in forests and scrublands. The transition from forests and scrublands into human development areas is the usual LULCC pattern. Although both, deforestation and afforestation occurred in the area, large-scale development of fast-growing trees appears to dominate forest restoration, i.e. the creation of lower quality ecosystems. Sustainable forest management should emphasize that mitigation of forest fragmentation is needed. According to the classification calculation, most of the forest changes take place infrastructure development. Changes in forested areas were very high near the pipelines, but this dropped off to virtually nil at the edge of the study area, indicating a linear relationship between forest loss and the construction of pipelines.

It can be concluded that over the two-year study period, the LULC rate of change, gains and losses as well as transfer rates are all high, suggesting that the LULC transition is intense along the pipelines given that all changes are related to infrastructure development. The LULCC results from Myanmar offer useful insights for other countries with oil and gas pipelines and transboundary infrastructure. The development of transnational energy projects triggers significant human and environmental security issues throughout the region. Although the discovery of new energy resources can be beneficial, the risks of serious ecosystem degradation from exploration and transportation of energy is high. Future research should assess the impacts of various scenarios of energy development on other environmental changes, such as water contamination, biodiversity depletion, and soil erosion, and harm to human health due to pipeline incidents. What will be of particular importance is to map not just total gains and losses of e.g. forested and agricultural areas, but the differential effects, for example with regards to the creation of lower quality forests.

## Supporting information

**S1 File. Landsat scenes and tiles used in the analysis.**
(DOCX)

**S2 File. R script used to classify GeoEye1 and worldview2 images are provided in the supporting information files.**
(DOCX)

## Author Contributions

**Conceptualization:** Thiri Shwesin Aung.

**Data curation:** Thiri Shwesin Aung.

**Formal analysis:** Thiri Shwesin Aung.

**Investigation:** Thiri Shwesin Aung.

**Methodology:** Thiri Shwesin Aung.

**Project administration:** Thiri Shwesin Aung.

**Resources:** Thiri Shwesin Aung.

**Software:** Thiri Shwesin Aung.

**Validation:** Thiri Shwesin Aung.

**Visualization:** Thiri Shwesin Aung.

**Writing – original draft:** Thiri Shwesin Aung.

**Writing – review & editing:** Thomas B. Fischer, John Buchanan.

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
