## [Decision Letter · Decision Letter 0]

10 Jun 2020

PONE-D-20-08091

Land Use and Land Cover Changes along the China-Myanmar Oil and Gas Pipelines – Monitoring Infrastructure Development in Remote Conflict-prone Regions

PLOS ONE

Dear Dr. Aung,

Thank you for submitting your manuscript to PLOS ONE. After careful consideration, we feel that it has merit but does not fully meet PLOS ONE’s publication criteria as it currently stands. Therefore, we invite you to submit a revised version of the manuscript that addresses the points raised during the review process.

While the analysis presented in this manuscript is fairly sound, both reviewers, but particularly Reviewer 2, make strong suggestions that would greatly enhance the manuscript. At this point, please reflect on, and address, all of Reviewer 2's comments, particularly the questions regarding the use of multiple sensing platforms and data types (i.e., why not include hyperspectral data with multispectral data for enhanced classification accuracy?). I am particularly interested to see your response to this suggestion. Finally, please also include line-numbers in your revised manuscript, which will help when we reevaluate the revised manuscript..

We look forward to receiving your revised manuscript.

Kind regards,

Stephen P. Aldrich, PhD

Academic Editor

PLOS ONE

Journal Requirements:

3. We note that Figures 1,2 and 3 in your submission contain [map/satellite] images which may be copyrighted. All PLOS content is published under the Creative Commons Attribution License (CC BY 4.0), which means that the manuscript, images, and Supporting Information files will be freely available online, and any third party is permitted to access, download, copy, distribute, and use these materials in any way, even commercially, with proper attribution. For these reasons, we cannot publish previously copyrighted maps or satellite images created using proprietary data, such as Google software (Google Maps, Street View, and Earth). For more information, see our copyright guidelines: http://journals.plos.org/plosone/s/licenses-and-copyright.

1.    You may seek permission from the original copyright holder of Figures 1,2 and 3 to publish the content specifically under the CC BY 4.0 license.

Additional Editor Comments (if provided):

While the analysis presented in this manuscript is fairly sound, both reviewers, but particularly Reviewer 2, make strong suggestions that would greatly enhance the manuscript. At this point, please reflect on, and address, all of Reviewer 2's comments, particularly the questions regarding the use of multiple sensing platforms and data types (i.e., why not include hyperspectral data with multispectral data for enhanced classification accuracy?). Please also include line-numbers in your revised manuscript.

Reviewers' comments:

Reviewer's Responses to Questions

**Comments to the Author**

1. Is the manuscript technically sound, and do the data support the conclusions?

Reviewer #1: Yes

Reviewer #2: Partly

2. Has the statistical analysis been performed appropriately and rigorously? 

Reviewer #1: Yes

Reviewer #2: Yes

3. Have the authors made all data underlying the findings in their manuscript fully available?

Reviewer #1: Yes

Reviewer #2: Yes

4. Is the manuscript presented in an intelligible fashion and written in standard English?

Reviewer #1: Yes

Reviewer #2: Yes

5. Review Comments to the Author

Reviewer #1: This study quantifies the gains and losses of different land uses along an oil pipeline in Myanmar, focused on forest. The study is based on remote sensed analysis with field validation in some sites.

The study is very well written and the study area, methods and results are described in detail. Discussions are also Ok, while figures and tables are of good quality.

I would accept the manuscript with a minor revision. The also made some visits to specific sites along the pipeline to verify location and land cover conditions. It would be nice to document these visits with some relevant photos (e.g. pipe line corridor, land cover destruction, the referred afforestation, etc).

Reviewer #2: Dear authors,

Your study investigated land use and land cover changes (LULCC) associated with the construction of the China-Myanmar oil and gas pipelines, specifically focusing at a study site in Rakhine State in Myanmar. The paper specifically aimed to address the following research questions: (1) what is the rate and pattern of LULCC along the pipelines? (2) how much forests have been lost during the study period? (3) what is the pattern of afforestation in the study area?

It is an interesting and relevant study that provided spatially explicit and quantifiable evidence of the land cover changes associated with the construction of the oil and gas pipelines traversing Myanmar leading to China through the classification and interpretation of remotely sensed satellite data. My major comments, detailed below, are primarily about improving the methods, specifically the sampling design and accuracy assessments, and being more thorough and comprehensive in citing the relevant previous studies on land use and land cover changes in Myanmar. Please find my comments below to improve this current version of your manuscript. Also, in your subsequent revised submissions, please provide line numbers and an outline of the different sections of the manuscript to make it easier for reviewers to provide feedback to specific parts of your manuscript.

INTRODUCTION

The Introduction can be improved and should be more comprehensive in its treatment of the relevant literature. Currently, there are several relevant studies on land use and land cover changes, including forest change, in Myanmar that are not referred to in the Introduction. For example:

• Page 3, Para 3 of Introduction: On environmental concerns about oil and gas pipelines  In addition to the examples in the Niger Delta or in Pennsylvania in the United States, the authors should mention examples from studies in Myanmar, particularly environmental impacts of another oil and gas pipeline in Myanmar, such as the Yetagon and Yadana pipeline (e.g., EarthRights International 2000, 2009).

• Page 4, last paragraph of Introduction: ‘…the country is suffering from significant annual forest loss due to infrastructure development, firewood overexploitation, illegal logging, shifting cultivation, and expansion of agricultural lands.’  What is the source of these causes and drivers of forest loss? The list of causes and drivers of forest loss should be more nuanced. Good examples are the studies by Lim et al (2017) and Prescott et al (2017) that identified drivers of deforestation and forest degradation in Myanmar.

Also in Page 4, last paragraph of Introduction:

• ‘Myanmar possessing largest remaining forest areas in Asia with over 40%... intact forest cover’  Citation is required here.

• Aside from FAO, cite studies that quantified forest cover and forest change in Myanmar, particularly Leimgruber et al (2005) and Bhagwat et al (2017)

Arakan Oil Watch (2009) is cited in the main text but not found in the references list.

STUDY REGION

What types/kinds of extensive anthropogenic activities have led to the decrease in forests in Ann Township or in Kyaukpyu District? Please be more specific and cite references as much as possible.

Page 4, Para 2 of Study Region: Instead of referring to lands that have never been cultivated as ‘virgin’ lands, I suggest plainly stating that some lands remain in pristine original condition and have not undergone cultivation. Also, cite the source/s of the land cover statistics of the study area described in Para 1 and Para 2 (e.g., 36.56% forest; 53% ‘uncultivated’ land; 1% rural, urban, industrial).

Page 5, Para 3 of Study Region:

• ‘Environmental degradation in the region is primarily linked to recent development projects, namely significant infrastructure developments and investment projects.’  What is the basis of this statement?

• Also, ‘Both pipelines transverse through the entire Rakhine State, Magwe Division, Mandalay Division, and Shan State before entering China.’  The authors mentioned that the oil and gas pipelines traverse through Myanmar (specifically Rakhine State, Magwe Region, Mandalay Region, and Shan State) to Yunnan Province in China. What is the source of this information?

DATA AND IMAGE PROCESSING

Page 8, Para 1 of Data and Image Processing: In this study, the authors primarily chose Landsat multispectral sensors for the land cover classification. Why did the authors not use a combination of satellite data from multiple sensors (such as multispectral, radar, and/or hyperspectral data) to generate their land use and land cover change analyses? This is an approach that has been demonstrated to yield better classification accuracies, particularly for land use/cover mapping and change analyses and for improved discrimination of land cover classes. In fact, several studies specifically in Myanmar have demonstrated improved classification accuracies using combined satellite sensor data such as by Torbick et al (2016), De Alban et al (2018, 2020), and Nomura et al (2019) compared to using image data from single sensors only.

Page 8-10, Para 2 and Para 4 and Para 6 of Data and Image Processing: While the authors cite Olofsson et al (2014), there are several missing elements of the accuracy assessments that the authors neglect to report in their study. The authors should discuss their sampling design in more detail based on Olofsson et al (2014). For example, what is the minimum number of samples required to achieve their target accuracy for each time-period or year based on their chosen sampling design? The authors should also report unbiased accuracies with confidence intervals for their land cover classification results. In addition to reporting the standard accuracy assessment metrics (OA, UA, PA), the error matrices should also be reported in terms of sample counts and estimated area proportions.

Table 4 indeed reports the error matrices for 2010 and 2012 in terms of sample counts but is presented in a confusing manner. Table 4 should be split into two matrices, one for 2010 and one for 2012, instead of presented together.

Page 9, Para 3 of Data and Image Processing:

• Cite/mention the sources of the GeoEye and WorldView images.

• ‘Image fusion was performed using the Gram-Schmidt Pan Sharpening method.’  This sentence is unclear. Was the pansharpening implemented on the Landsat images by fusing them with the high-resolution GeoEye and WorldView images?

Page 9, Para 4 of Data and Image Processing: Enumerate the predictor variables that were used for the land cover classification and how many variables (total) were used.

Page 10, Para 5 of Data and Image Processing:

• ‘The model also calculates gains and losses, as well as project net changes, and determines drivers of change for each land cover category, both, in map and graphical form.’  How does the LCM model determine the drivers of the changes for each land cover category?

RESULTS

I suggest moving the Accuracy Assessment section before the Land Use and Land Cover Change Along the Pipelines section. Presenting the results of the accuracy assessments of the land cover classification first give readers a sense of whether the resulting land cover maps are reliable (or not) for subsequent change analysis.

Section on Land Use and Land Cover Change Along the Pipelines: It is not clear how much was the buffer distance used to calculate the land cover changes attributed to the construction of the pipelines. How much buffer distance was used in the calculation and how was the buffer distance selected?

Page 14, Para 1 of Accuracy Assessment:

• ‘This relatively lower overall accuracy can be due to the use of a small study area and / or small training samples.’  This statement is difficult to substantiate without a sampling design. Hence the need to describe the sampling design used for the study in the Methods section (see earlier comment above).

• I do not think the low accuracies of Forest and Scrubland can be attributed to ‘…the more significant number of small patches associated with both land cover types,’ as the authors claim. Based on the land cover maps, Forest is the largest and most extensive land cover class, hence it is definitely not due to ‘small patches’ of Forest that is driving its low accuracy. Please delete this sentence.

• The answer to the low accuracy of Forest and Scrubland can be found by inspecting the error matrices in Table 4, which indicated that the Random Forest algorithm was ‘confused’ in distinguishing between Forest and Scrubland (e.g., 461 and 428 pixels of Scrubland were misclassified as Forest in 2010 and 2012, respectively; 1429 and 1418 pixels of Forest were misclassified as Forest in 2010 and 2012, respectively), given that both land cover classes exhibit similar vegetation characteristics (which may also be due to the samples used for the classification). And this is what the authors indeed say in the subsequent sentences: ‘Forests might sometimes be assigned falsely as scrublands and vice versa during the training digitization process due to their visual similarities, leading to either over- or underestimation,’ to which I agree.

• ‘Our accuracy values are higher than those reported in the previous study, conducted for countrywide forest cover changes in Myanmar, where overall forest accuracy for small patches of the forest was 50% (Bhagwat et al., 2017).’  Delete this sentence. The accuracies reported in this study is not comparable to the Bhagwat et al (2017) study due to difference in geographic scales.

DISCUSSION

Page 20, Para 3 of Discussion: In addition to citing Borneo as an example of commercial plantation expansion replacing forests, there are several examples from studies specifically in Myanmar that should be mentioned and cited, particularly evidence found in Tanintharyi Region (e.g., Woods (2015); De Alban et al (2018); Woods (2019); De Alban et al (2019); Zaehringer et al (2018); Zaehringer et al (2020)).

Page 20, Para 4 of Discussion: ‘As can be expected, areas closer to infrastructure development tended to experience higher rates of transitions from forests, scrublands and agricultural land to other land use types, while further away areas were less likely to convert from tree-covered into built-up areas.’  This has not been tested explicitly in this study. I suggest revising this sentence.

CONCLUSIONS

Page 22, Para 2 of Conclusion: These research questions should be presented in the Introduction of the study (currently it is not in the Introduction) and then revisited in the Conclusion.

FIGURES AND TABLES

Refer to all the figures in the main text, where appropriate. For example, when the study area is described, there is no reference to Figure 1. Presently, only Figures 3, 6, and 7 are referred to in the main text.

The numbering of tables should be in order. For example, currently, Table 4 is presented before Table 2.

Map scales units should be consistent for all map figures. For example, Figure 2 scale units is in miles; Figure 3 scale units is in kilometers. The authors could either adopt a singular scale unit for consistency, or present both scale units (thus two scale bars in each figure, one in miles and one in kilometers).

Comments for Figure 1:

• Figure 1 can be improved to allow readers to better appreciate the location of the study area, these administrative areas, and the extent traversed by the pipeline that were described in the Study Region section. Since the study deals with the oil and gas pipeline, the location of the pipelines traversing through Myanmar all the way to Yunnan Province in China should be shown, similar to the inset in Figure 3. The Myanmar states/regions traversed by the pipeline should then be labeled (i.e., Rakhine, Magway, Mandalay, Shan) as well as Yunnan, China as these locations are specifically mentioned in the text.

• Also, in Figure 1, the map figure should also present the study area, specifically Kyaukpyu District of Rakhine State, on a much larger scale compared to how it is currently presented, which is too small. It is also difficult to distinguish the various townships in the legend against the map of Rakhine State due to both the choice of color scheme and small-scale depiction of Rakhine State. Instead, Rakhine State should be presented on a larger scale, its townships labeled instead of presented as a separate legend. The labels in the satellite image inset, including the red polygon in the satellite image, as well as the scale bar at the bottom of the map are too small to see or read. The font sizes and scale bar should be increased to make the text clear and readable. The north arrow is misplaced near the legend and is hardly noticeable; the arrow can either be moved elsewhere to make it visible or removed. Grids and graticules should also be present along the border of the map.

In the caption of Figure 2, indicate which satellite sensor and the band numbers of the RGB composites that were used for each year.

In Figure 3, are the false color composites generated from Landsat? GeoEye? WorldView? Also, in Figure 3 caption, I suggest stating which bands were used to show the false color composites. What is the purpose of this figure? Delete? Or combine with Figure 1?

In Figure 4, it is not easy to distinguish Agriculture, Infrastructure Development, and Residential Area in the land cover maps. Perhaps the color scheme can be improved.

For Figure 5, the land area units (in km2?) should be stated, either in the caption or the figure itself. Table 2 is redundant and should be deleted since the information is presented already in Figure 5. Change the color of the bars for net change (bottom plot) to differentiate it from the color of the bars depicting losses in the gross gains and losses (top plot).

For Figure 6, change the legend title to ‘Land Cover Change’ instead of ‘Land Cover Classification’ as the information presented in the map are the changes or transitions from one land cover class to another.

REFERENCES

1. EarthRights International. Total Denial Continues: Earth Rights Abuses Along the Yadana and Yetagun Pipelines in Burma. Washington, DC, USA: EarthRights International; 2000 May p. 183. Available: https://earthrights.org/wp-content/uploads/publications/Total-Denial-Continues-2000.pdf

2. EarthRights International. Total Impact: The Human Rights, Environmental, and Financial Impacts of Total and Chevron’s Yadana Gas Project in Military-Ruled Burma (Myanmar). Washington, DC, USA: EarthRights International; 2009 Sep p. 107. Available: https://earthrights.org/wp-content/uploads/publications/total-impact.pdf

3. Lim CL, Prescott GW, De Alban JDT, Ziegler AD, Webb EL. Untangling the proximate causes and underlying drivers of deforestation and forest degradation in Myanmar. Conserv Biol. 2017;31: 1362–1372. doi:10.1111/cobi.12984

4. Prescott GW, Sutherland WJ, Aguirre D, Baird M, Bowman V, Brunner J, et al. Political transition and emergent forest-conservation issues in Myanmar. Conserv Biol. 2017;31: 1257–1270. doi:10.1111/cobi.13021

5. Torbick N, Ledoux L, Salas W, Zhao M. Regional mapping of plantation extent using multisensor imagery. Remote Sens. 2016;8: 236. doi:10.3390/rs8030236

6. De Alban JDT, Connette GM, Oswald P, Webb EL. Combined Landsat and L-Band SAR data improves land cover classification and change detection in dynamic tropical landscapes. Remote Sens. 2018;10: 306. doi:10.3390/rs10020306

7. De Alban JDT, Jamaludin J, Wen DW de, Than MM, Webb EL. Improved estimates of mangrove cover and change reveal catastrophic deforestation in Myanmar. Environ Res Lett. 2020;15: 034034. doi:10.1088/1748-9326/ab666d

8. Nomura K, Mitchard ETA, Patenaude G, Bastide J, Oswald P, Nwe T. Oil palm concessions in southern Myanmar consist mostly of unconverted forest. Sci Rep. 2019;9: 1–9. doi:10.1038/s41598-019-48443-3

9. Woods K. Commercial Agriculture Expansion in Myanmar: Links to Deforestation, Conversion Timber, and Land Conflicts. Washington, DC, USA: Forest Trends and UKAID; 2015 Mar p. 78.

10. Woods KM. Green territoriality: conservation as state territorialization in a resource frontier. Hum Ecol. 2019 [cited 12 Mar 2019]. doi:10.1007/s10745-019-0063-x

11. De Alban JDT, Prescott GW, Woods KM, Jamaludin J, Latt KT, Lim CL, et al. Integrating analytical frameworks to investigate land-cover regime shifts in dynamic landscapes. Sustainability. 2019;11: 1139. doi:10.3390/su11041139

12. Zaehringer JG, Llopis JC, Latthachack P, Thein TT, Heinimann A. A novel participatory and remote-sensing-based approach to mapping annual land use change on forest frontiers in Laos, Myanmar, and Madagascar. J Land Use Sci. 2018;0: 1–16. doi:10.1080/1747423X.2018.1447033

13. Zaehringer JG, Lundsgaard-Hansen L, Thein TT, Llopis JC, Tun NN, Myint W, et al. The cash crop boom in southern Myanmar: tracing land use regime shifts through participatory mapping. Ecosyst People. 2020;16: 36–49. doi:10.1080/26395916.2019.1699164

6. PLOS authors have the option to publish the peer review history of their article (what does this mean?). If published, this will include your full peer review and any attached files.

Reviewer #1: Yes: Fernando A.L. Pacheco

Reviewer #2: No

---

## [Author Response · Author response to Decision Letter 0]

26 Jun 2020

Thank you for giving us the opportunity to submit a revision of our manuscript. We truly appreciate all the constructive comments and suggestions from the reviewers. 

We have adopted all the suggestions and revised each sections of our manuscript. The following are our point-to-point responses to the reviewers’ comments. 

Review Comments to the Author

Reviewer #1: This study quantifies the gains and losses of different land uses along an oil pipeline in Myanmar, focused on forest. The study is based on remote sensed analysis with field validation in some sites.

The study is very well written and the study area, methods and results are described in detail. Discussions are also Ok, while figures and tables are of good quality.

I would accept the manuscript with a minor revision. The also made some visits to specific sites along the pipeline to verify location and land cover conditions. It would be nice to document these visits with some relevant photos (e.g. pipe line corridor, land cover destruction, the referred afforestation, etc).

Response: Thank you for this important suggestion. We have included some of the photos from our filed work showing pipelines crossing the forested areas. 

Reviewer #2: Dear authors,

Your study investigated land use and land cover changes (LULCC) associated with the construction of the China-Myanmar oil and gas pipelines, specifically focusing at a study site in Rakhine State in Myanmar. The paper specifically aimed to address the following research questions: (1) what is the rate and pattern of LULCC along the pipelines? (2) how much forests have been lost during the study period? (3) what is the pattern of afforestation in the study area?

It is an interesting and relevant study that provided spatially explicit and quantifiable evidence of the land cover changes associated with the construction of the oil and gas pipelines traversing Myanmar leading to China through the classification and interpretation of remotely sensed satellite data. My major comments, detailed below, are primarily about improving the methods, specifically the sampling design and accuracy assessments, and being more thorough and comprehensive in citing the relevant previous studies on land use and land cover changes in Myanmar. Please find my comments below to improve this current version of your manuscript. Also, in your subsequent revised submissions, please provide line numbers and an outline of the different sections of the manuscript to make it easier for reviewers to provide feedback to specific parts of your manuscript.

INTRODUCTION

1. The Introduction can be improved and should be more comprehensive in its treatment of the relevant literature. Currently, there are several relevant studies on land use and land cover changes, including forest change, in Myanmar that are not referred to in the Introduction. For example:

• Page 3, Para 3 of Introduction: On environmental concerns about oil and gas pipelines  In addition to the examples in the Niger Delta or in Pennsylvania in the United States, the authors should mention examples from studies in Myanmar, particularly environmental impacts of another oil and gas pipeline in Myanmar, such as the Yetagon and Yadana pipeline (e.g., EarthRights International 2000, 2009).

• Page 4, last paragraph of Introduction: ‘…the country is suffering from significant annual forest loss due to infrastructure development, firewood overexploitation, illegal logging, shifting cultivation, and expansion of agricultural lands.’  What is the source of these causes and drivers of forest loss? The list of causes and drivers of forest loss should be more nuanced. Good examples are the studies by Lim et al (2017) and Prescott et al (2017) that identified drivers of deforestation and forest degradation in Myanmar.

Also in Page 4, last paragraph of Introduction:

• ‘Myanmar possessing largest remaining forest areas in Asia with over 40%... intact forest cover’  Citation is required here.

• Aside from FAO, cite studies that quantified forest cover and forest change in Myanmar, particularly Leimgruber et al (2005) and Bhagwat et al (2017)

Arakan Oil Watch (2009) is cited in the main text but not found in the references list.

Response: Thank you for suggesting important references for this paper. We have cited the suggested references and other relevant literatures in the introduction section. Please see Page 3 &4. 

STUDY REGION

2. What types/kinds of extensive anthropogenic activities have led to the decrease in forests in Ann Township or in Kyaukpyu District? Please be more specific and cite references as much as possible.

Response: Page 4_We have specified the anthropogenic activities and cited reference paper. 

Page 4, Para 2 of Study Region: Instead of referring to lands that have never been cultivated as ‘virgin’ lands, I suggest plainly stating that some lands remain in pristine original condition and have not undergone cultivation. Also, cite the source/s of the land cover statistics of the study area described in Para 1 and Para 2 (e.g., 36.56% forest; 53% ‘uncultivated’ land; 1% rural, urban, industrial).

Response: Thank you for the comment. We have revised the sentence and added citation where appropriate (Page 4). 

Page 5, Para 3 of Study Region:

• ‘Environmental degradation in the region is primarily linked to recent development projects, namely significant infrastructure developments and investment projects.’  What is the basis of this statement?

• Also, ‘Both pipelines transverse through the entire Rakhine State, Magwe Division, Mandalay Division, and Shan State before entering China.’  The authors mentioned that the oil and gas pipelines traverse through Myanmar (specifically Rakhine State, Magwe Region, Mandalay Region, and Shan State) to Yunnan Province in China. What is the source of this information?

Response: Thank you for pointing this out. Due to technical error, some of the references we cited earlier have disappeared automatically. We have added them again where appropriate. Please see Page 5. 

DATA AND IMAGE PROCESSING

3. Page 8, Para 1 of Data and Image Processing: In this study, the authors primarily chose Landsat multispectral sensors for the land cover classification. Why did the authors not use a combination of satellite data from multiple sensors (such as multispectral, radar, and/or hyperspectral data) to generate their land use and land cover change analyses? This is an approach that has been demonstrated to yield better classification accuracies, particularly for land use/cover mapping and change analyses and for improved discrimination of land cover classes. In fact, several studies specifically in Myanmar have demonstrated improved classification accuracies using combined satellite sensor data such as by Torbick et al (2016), De Alban et al (2018, 2020), and Nomura et al (2019) compared to using image data from single sensors only.

Response: Thank you for this comment. Firstly, we did not use Landsat multispectral sensors for the land cover classification. We used we Landsat 5, 7, and 8 to validate the pipeline route and the year of construction before purchasing VHRI images. The main satellite data used for the classification of Land-Use-Land-Cover-Change (LULCC) are commercial VHRI orthorectified multispectral satellite images, GeoEye-1 and Worldview-2. The need to purchase very expensive satellite images were required due to lack of freely downloadable high-resolution images in the study area during the study period. 

For the use of hyperspectral imageries, there was no airborne hyperspectral imagery for the area for 2010-2012. The earliest available year was 2017. In addition, hyperspectral imageries have spatial resolution too coarse for our mapping purposes. They are particularly useful to detect finer features such as tree species within the forest, pollutant, hazardous waste etc., which is out of scope of our study.

The satellites used in the study are two of the world’s highest resolution commercial earth-imaging satellites and have 0.5m resolution. Previous studies combined multisensors when using low to medium resolution imageries to compensate limitations of each satellite. As VHRI already offer highest resolution available, we do not think that combining them with lower resolution images will produce any useful results. 

To clarify the methodology, we further explained this in the manuscript. Please see Page 8. 

4. Page 8-10, Para 2 and Para 4 and Para 6 of Data and Image Processing: While the authors cite Olofsson et al (2014), there are several missing elements of the accuracy assessments that the authors neglect to report in their study. The authors should discuss their sampling design in more detail based on Olofsson et al (2014). For example, what is the minimum number of samples required to achieve their target accuracy for each time-period or year based on their chosen sampling design? The authors should also report unbiased accuracies with confidence intervals for their land cover classification results. In addition to reporting the standard accuracy assessment metrics (OA, UA, PA), the error matrices should also be reported in terms of sample counts and estimated area proportions.Table 4 indeed reports the error matrices for 2010 and 2012 in terms of sample counts but is presented in a confusing manner. Table 4 should be split into two matrices, one for 2010 and one for 2012, instead of presented together.

Response: Thank you for this comment. Based on the suggestion, we have divided the confusion matrix table for 2010 and 2012. We also reported confidence intervals. Please see Page 14 (line 367-368). In our study, we solely relied on validation approach for accuracy assessment in Random Forest algorithm without further analysis or manual calculation. Unlike OOB or ROC, validation score is computed by setting aside a part of the original dataset before training models. The samples used by RF model can be reflected in the confusion matrix. We added this explanation now in Page 10 & 11.

5. Page 9, Para 3 of Data and Image Processing:

• Cite/mention the sources of the GeoEye and WorldView images.

Response: Page 9_We now cited the source where we purchased the images. 

• ‘Image fusion was performed using the Gram-Schmidt Pan Sharpening method.’  This sentence is unclear. Was the pansharpening implemented on the Landsat images by fusing them with the high-resolution GeoEye and WorldView images?

Response: Paragraph 4 only refers to GeoEye and WorldView images as these are the data used for LULCC analysis. Landsat imageries were only used for pipelines route identification. We have clarified this now in the data section. 

6. Page 9, Para 4 of Data and Image Processing: Enumerate the predictor variables that were used for the land cover classification and how many variables (total) were used.

Response: Following this comment, we have explained the predictor variables used in RF model. 

7. Page 10, Para 5 of Data and Image Processing:

• ‘The model also calculates gains and losses, as well as project net changes, and determines drivers of change for each land cover category, both, in map and graphical form.’  How does the LCM model determine the drivers of the changes for each land cover category?

Response: By drivers of change, we meant land type associated with each conversion pattern as summarized in Table 3 (Page 17). 

RESULTS

8. I suggest moving the Accuracy Assessment section before the Land Use and Land Cover Change Along the Pipelines section. Presenting the results of the accuracy assessments of the land cover classification first give readers a sense of whether the resulting land cover maps are reliable (or not) for subsequent change analysis.

Response: Page 12_We have moved the Accuracy Assessment section before the Land Use and Land Cover Change Along the Pipelines section.

9. Section on Land Use and Land Cover Change Along the Pipelines: It is not clear how much was the buffer distance used to calculate the land cover changes attributed to the construction of the pipelines. How much buffer distance was used in the calculation and how was the buffer distance selected?

Response: We did not use any buffer zone in the calculation of LULCC as we have already included pipelines ROW in the analysis. To clarify, we added this explanation in “study region” section (page 5). 

Page 14, Para 1 of Accuracy Assessment:

• ‘This relatively lower overall accuracy can be due to the use of a small study area and / or small training samples.’  This statement is difficult to substantiate without a sampling design. Hence the need to describe the sampling design used for the study in the Methods section (see earlier comment above).

Response: “Small training samples” refers to 370 training polygons identified in Google Earth Pro and used in RF model. Please see Page 9. 

• I do not think the low accuracies of Forest and Scrubland can be attributed to ‘…the more significant number of small patches associated with both land cover types,’ as the authors claim. Based on the land cover maps, Forest is the largest and most extensive land cover class, hence it is definitely not due to ‘small patches’ of Forest that is driving its low accuracy. Please delete this sentence. The answer to the low accuracy of Forest and Scrubland can be found by inspecting the error matrices in Table 4, which indicated that the Random Forest algorithm was ‘confused’ in distinguishing between Forest and Scrubland (e.g., 461 and 428 pixels of Scrubland were misclassified as Forest in 2010 and 2012, respectively; 1429 and 1418 pixels of Forest were misclassified as Forest in 2010 and 2012, respectively), given that both land cover classes exhibit similar vegetation characteristics (which may also be due to the samples used for the classification). And this is what the authors indeed say in the subsequent sentences: ‘Forests might sometimes be assigned falsely as scrublands and vice versa during the training digitization process due to their visual similarities, leading to either over- or underestimation,’ to which I agree.

Response: Based on this comment, we have deleted the sentence. Please see Page 12. 

• ‘Our accuracy values are higher than those reported in the previous study, conducted for countrywide forest cover changes in Myanmar, where overall forest accuracy for small patches of the forest was 50% (Bhagwat et al., 2017).’  Delete this sentence. The accuracies reported in this study is not comparable to the Bhagwat et al (2017) study due to difference in geographic scales.

Response: Based on this comment, we have deleted the sentence. Please see Page 12. 

DISCUSSION

Page 20, Para 3 of Discussion: In addition to citing Borneo as an example of commercial plantation expansion replacing forests, there are several examples from studies specifically in Myanmar that should be mentioned and cited, particularly evidence found in Tanintharyi Region (e.g., Woods (2015); De Alban et al (2018); Woods (2019); De Alban et al (2019); Zaehringer et al (2018); Zaehringer et al (2020)).

Response: Thank you for suggesting these important references. We have cited them where appropriate. Please see Page 20 & 21. 

Page 20, Para 4 of Discussion: ‘As can be expected, areas closer to infrastructure development tended to experience higher rates of transitions from forests, scrublands and agricultural land to other land use types, while further away areas were less likely to convert from tree-covered into built-up areas.’  This has not been tested explicitly in this study. I suggest revising this sentence.

Response: This discussion refers to the results from Figure 7. We have revised the sentences to clarify this. 

CONCLUSIONS

Page 22, Para 2 of Conclusion: These research questions should be presented research questions of the study (currently it is not in the Introduction) and then revisited in the Conclusion.

Response: Following this suggestion, we have added the research questions in the introduction section. 

FIGURES AND TABLES

Refer to all the figures in the main text, where appropriate. For example, when the study area is described, there is no reference to Figure 1. Presently, only Figures 3, 6, and 7 are referred to in the main text.

Response: We added references for each figures. 

The numbering of tables should be in order. For example, currently, Table 4 is presented before Table 2.

Response: We corrected the order of tables. 

Map scales units should be consistent for all map figures. For example, Figure 2 scale units is in miles; Figure 3 scale units is in kilometers. The authors could either adopt a singular scale unit for consistency, or present both scale units (thus two scale bars in each figure, one in miles and one in kilometers).

Response: We edited the scale bar of Figure 3. 

Comments for Figure 1:

• Figure 1 can be improved to allow readers to better appreciate the location of the study area, these administrative areas, and the extent traversed by the pipeline that were described in the Study Region section. Since the study deals with the oil and gas pipeline, the location of the pipelines traversing through Myanmar all the way to Yunnan Province in China should be shown, similar to the inset in Figure 3. The Myanmar states/regions traversed by the pipeline should then be labeled (i.e., Rakhine, Magway, Mandalay, Shan) as well as Yunnan, China as these locations are specifically mentioned in the text.

• Also, in Figure 1, the map figure should also present the study area, specifically Kyaukpyu District of Rakhine State, on a much larger scale compared to how it is currently presented, which is too small. It is also difficult to distinguish the various townships in the legend against the map of Rakhine State due to both the choice of color scheme and small-scale depiction of Rakhine State. Instead, Rakhine State should be presented on a larger scale, its townships labeled instead of presented as a separate legend. The labels in the satellite image inset, including the red polygon in the satellite image, as well as the scale bar at the bottom of the map are too small to see or read. The font sizes and scale bar should be increased to make the text clear and readable. The north arrow is misplaced near the legend and is hardly noticeable; the arrow can either be moved elsewhere to make it visible or removed. Grids and graticules should also be present along the border of the map.

Response: Thank you for this important suggestion. Following this comment, we have added a new study area map (page 5). We included pipeline route as in Figure 3. We were required to remove the image showing pipeline polygon on the map as the journal does not allow using Google Earth imagery. As our country of interest is Myanmar and we do not have exact data for pipeline route in China, we decided not to complicate the map by including Yunnan in the study area map. 

In the caption of Figure 3, indicate which satellite sensor and the band numbers of the RGB composites that were used for each year.

Response: Page 11_We have changed the scale bar to miles and indicated data and RGB composites used for each year. 

In Figure 3, are the false color composites generated from Landsat? GeoEye? WorldView? Also, in Figure 3 caption, I suggest stating which bands were used to show the false color composites. What is the purpose of this figure? Delete? Or combine with Figure 1?

Response: Figure 3 display how the pipelines route have been identified based on Landsat data before LULCC classification using GeoEye1 and WorldView2. Please see page 8 & 11. 

In Figure 4, it is not easy to distinguish Agriculture, Infrastructure Development, and Residential Area in the land cover maps. Perhaps the color scheme can be improved.

Response: Page 14_ We have reproduced the images in ArcGis Pro instead of ArcMap. The resolution of the maps significantly improved. We have also chosen different colors scheme. 

For Figure 5, the land area units (in km2?) should be stated, either in the caption or the figure itself. Table 2 is redundant and should be deleted since the information is presented already in Figure 5. Change the color of the bars for net change (bottom plot) to differentiate it from the color of the bars depicting losses in the gross gains and losses (top plot).

Response: As Table 2 gives values and more useful information than the figure. We decided to delete Figure 5 and leave Table 2 instead. 

For Figure 6, change the legend title to ‘Land Cover Change’ instead of ‘Land Cover Classification’ as the information presented in the map are the changes or transitions from one land cover class to another.

Response: Page 17_We have changed the title as suggested. 

REFERENCES

1. EarthRights International. Total Denial Continues: Earth Rights Abuses Along the Yadana and Yetagun Pipelines in Burma. Washington, DC, USA: EarthRights International; 2000 May p. 183. Available: https://earthrights.org/wp-content/uploads/publications/Total-Denial-Continues-2000.pdf

2. EarthRights International. Total Impact: The Human Rights, Environmental, and Financial Impacts of Total and Chevron’s Yadana Gas Project in Military-Ruled Burma (Myanmar). Washington, DC, USA: EarthRights International; 2009 Sep p. 107. Available: https://earthrights.org/wp-content/uploads/publications/total-impact.pdf

3. Lim CL, Prescott GW, De Alban JDT, Ziegler AD, Webb EL. Untangling the proximate causes and underlying drivers of deforestation and forest degradation in Myanmar. Conserv Biol. 2017;31: 1362–1372. doi:10.1111/cobi.12984

4. Prescott GW, Sutherland WJ, Aguirre D, Baird M, Bowman V, Brunner J, et al. Political transition and emergent forest-conservation issues in Myanmar. Conserv Biol. 2017;31: 1257–1270. doi:10.1111/cobi.13021

5. Torbick N, Ledoux L, Salas W, Zhao M. Regional mapping of plantation extent using multisensor imagery. Remote Sens. 2016;8: 236. doi:10.3390/rs8030236

6. De Alban JDT, Connette GM, Oswald P, Webb EL. Combined Landsat and L-Band SAR data improves land cover classification and change detection in dynamic tropical landscapes. Remote Sens. 2018;10: 306. doi:10.3390/rs10020306

7. De Alban JDT, Jamaludin J, Wen DW de, Than MM, Webb EL. Improved estimates of mangrove cover and change reveal catastrophic deforestation in Myanmar. Environ Res Lett. 2020;15: 034034. doi:10.1088/1748-9326/ab666d

8. Nomura K, Mitchard ETA, Patenaude G, Bastide J, Oswald P, Nwe T. Oil palm concessions in southern Myanmar consist mostly of unconverted forest. Sci Rep. 2019;9: 1–9. doi:10.1038/s41598-019-48443-3

9. Woods K. Commercial Agriculture Expansion in Myanmar: Links to Deforestation, Conversion Timber, and Land Conflicts. Washington, DC, USA: Forest Trends and UKAID; 2015 Mar p. 78.

10. Woods KM. Green territoriality: conservation as state territorialization in a resource frontier. Hum Ecol. 2019 [cited 12 Mar 2019]. doi:10.1007/s10745-019-0063-x

11. De Alban JDT, Prescott GW, Woods KM, Jamaludin J, Latt KT, Lim CL, et al. Integrating analytical frameworks to investigate land-cover regime shifts in dynamic landscapes. Sustainability. 2019;11: 1139. doi:10.3390/su11041139

12. Zaehringer JG, Llopis JC, Latthachack P, Thein TT, Heinimann A. A novel participatory and remote-sensing-based approach to mapping annual land use change on forest frontiers in Laos, Myanmar, and Madagascar. J Land Use Sci. 2018;0: 1–16. doi:10.1080/1747423X.2018.1447033

13. Zaehringer JG, Lundsgaard-Hansen L, Thein TT, Llopis JC, Tun NN, Myint W, et al. The cash crop boom in southern Myanmar: tracing land use regime shifts through participatory mapping. Ecosyst People. 2020;16: 36–49. doi:10.1080/26395916.2019.1699164

---

## [Decision Letter · Decision Letter 1]

23 Jul 2020

PONE-D-20-08091R1

Land Use and Land Cover Changes along the China-Myanmar Oil and Gas Pipelines – Monitoring Infrastructure Development in Remote Conflict-prone Regions

PLOS ONE

Dear Dr. Aung,

Thank you for submitting your manuscript to PLOS ONE. After careful consideration, we feel that it has merit but does not fully meet PLOS ONE’s publication criteria as it currently stands. Therefore, we invite you to submit a revised version of the manuscript that addresses the points raised during the review process.

Thank you very much for your quality revision to this manuscript, which has significantly improved it. However, before I can render an "Accept" decision, I do believe that the comments of Reviewer 2 should be more adequately addressed. Reviewer 2 has graciously indicated a number of sources, and tools, that can be used to more adequately address the issue of accuracy assessment, which is a substantial aspect of any land use and land cover change study. I agree with Reviewer 2 that the indicated edits don't appear to have been made (or at least made completely), and would advise you to consider bumping up the accuracy assessment aspects of the manuscript. If you decide not to do so then you will need to provide a thorough defense of that decision that rebuts Reviewer 2's suggestions in a way that would satisfy both Reviewer 2, and myself.

If you can make these changes I anticipate reviewing the manuscript myself, and soliciting comments from Reviewer 2 in order to render a recommendation to the journal.

We look forward to receiving your revised manuscript.

Kind regards,

Stephen P. Aldrich, PhD

Academic Editor

PLOS ONE

Additional Editor Comments (if provided):

Thank you very much for your quality revision to this manuscript, which has significantly improved it. However, before I can render an "Accept" decision, I do believe that the comments of Reviewer 2 should be more adequately addressed. Reviewer 2 has graciously indicated a number of sources, and tools, that can be used to more adequately address the issue of accuracy assessment, which is a substantial aspect of any land use and land cover change study. I agree with Reviewer 2 that the indicated edits don't appear to have been made (or at least made completely), and would advise you to consider bumping up the accuracy assessment aspects of the manuscript. If you decide not to do so then you will need to provide a thorough defense of that decision that rebuts Reviewer 2's suggestions in a way that would satisfy both Reviewer 2, and myself.

If you can make these changes I anticipate reviewing the manuscript myself, and soliciting comments from Reviewer 2 in order to render a recommendation to the journal.

Reviewers' comments:

Reviewer's Responses to Questions

**Comments to the Author**

1. If the authors have adequately addressed your comments raised in a previous round of review and you feel that this manuscript is now acceptable for publication, you may indicate that here to bypass the “Comments to the Author” section, enter your conflict of interest statement in the “Confidential to Editor” section, and submit your "Accept" recommendation.

Reviewer #1: All comments have been addressed

Reviewer #2: (No Response)

2. Is the manuscript technically sound, and do the data support the conclusions?

Reviewer #1: Yes

Reviewer #2: Partly

3. Has the statistical analysis been performed appropriately and rigorously? 

Reviewer #1: Yes

Reviewer #2: No

4. Have the authors made all data underlying the findings in their manuscript fully available?

Reviewer #1: Yes

Reviewer #2: Yes

5. Is the manuscript presented in an intelligible fashion and written in standard English?

Reviewer #1: Yes

Reviewer #2: Yes

6. Review Comments to the Author

Reviewer #1: No further comments. The authors have adequately addressed all comments and suggestions I provided with the original manuscript and therefore the revised version is prepared to be published in PLOS ONE

Reviewer #2: Dear authors,

Thank you for considering my comments and suggestions in the revision of your manuscript.

First, the revised version of your manuscript incorporated the relevant literature more thoroughly, at least the studies that mapped and quantified land use/cover change as well as identified the causes and drivers of those changes in Myanmar. The methods are also much clearer now. For example, in the Data and Image Processing section, the main satellite images that you used in your land cover classification and change analysis were the very high-resolution images from GeoEye-1 and WorldView-2, with the moderate spatial resolution Landsat optical/multispectral images as complementary satellite data to the very high-resolution ones. Also, the pan-sharpening step was applied to the very high-resolution images using both their multispectral bands and panchromatic band. The revisions to figures and tables are also good. The inclusion of line numbers also help—thank you for this.

However, I think there is still room to improve the manuscript, particularly in adhering to the good practice recommendations outlined by Olofsson et al (2014) for producing scientifically rigorous and transparent estimates of accuracy and area in land use/cover change studies. For instance, the sampling design used for the study is still unclear (e.g., did you adopt a simple random sampling, or stratified random sampling, or another sampling approach for the accuracy assessment?). What target accuracy (or accuracies) were you aiming for in the land cover maps based on your study objectives? Based on your chosen sampling design, what was the required sample size and how many training and testing samples were used? For the results of the accuracy assessments, you’ve responded that you have reported the accuracies (overall, user’s, and producer’s) with confidence intervals, but I do not see these reflected in Tables 1 and 2 (in commission/omission errors) or in L 357-377 in Accuracy Assessment section of Results. Also, the error matrices should be presented in terms of both sample counts and estimated area proportions. Currently, Tables 1 and 2 present the error matrices for land cover maps in 2010 and 2012 in terms of sample counts, but I do not see the error matrices presented yet in terms of estimated area proportions. Finally, Table 3 should then present the adjusted area estimates based on the results of the accuracy assessment.

Please refer more closely to the key reference by Olofsson et al (2014), as well as Stehman & Foody (2019); both papers provide the good practice methodology for rigorous and transparent accuracy assessment of land cover products. The authors may also refer to the tools for error estimation developed by Olofsson et al (see https://github.com/beeoda/tutorials/tree/master/4_Estimation and http://beeoda.org/), as well as some additional references that applied these recommended good practices for accuracy assessment and area estimation for land cover change specifically in Myanmar—e.g., De Alban et al (2018, 2020) and Nomura et al (2019)—all of which used the Random Forest algorithm for land cover classification similar to this current study.

References

1. Olofsson P, Foody GM, Herold M, Stehman SV, Woodcock CE, Wulder MA. Good practices for estimating area and assessing accuracy of land change. Remote Sens Environ. 2014;148: 42–57. doi:10.1016/j.rse.2014.02.015

2. Stehman SV, Foody GM. Key issues in rigorous accuracy assessment of land cover products. Remote Sens Environ. 2019;231: 111199. doi:10.1016/j.rse.2019.05.018

3. De Alban JDT, Connette GM, Oswald P, Webb EL. Combined Landsat and L-Band SAR data improves land cover classification and change detection in dynamic tropical landscapes. Remote Sens. 2018;10: 306. doi:10.3390/rs10020306

4. De Alban JDT, Jamaludin J, Wen DW de, Than MM, Webb EL. Improved estimates of mangrove cover and change reveal catastrophic deforestation in Myanmar. Environ Res Lett. 2020;15: 034034. doi:10.1088/1748-9326/ab666d

5. Nomura K, Mitchard ETA, Patenaude G, Bastide J, Oswald P, Nwe T. Oil palm concessions in southern Myanmar consist mostly of unconverted forest. Sci Rep. 2019;9: 1–9. doi:10.1038/s41598-019-48443-3

7. PLOS authors have the option to publish the peer review history of their article (what does this mean?). If published, this will include your full peer review and any attached files.

Reviewer #1: **Yes: **Fernando António Leal Pacheco

Reviewer #2: No

---

## [Author Response · Author response to Decision Letter 1]

27 Jul 2020

Thank you for giving us the opportunity to submit a revision of our manuscript. We truly appreciate all the constructive comments and suggestions from the reviewers. 

We have adopted all the suggestions and revised each sections of our manuscript. Following the suggestion, we have followed (Olofsson et al., 2014)’s method for accuracy assessment. The following are our point-to-point responses to the reviewers’ comments. 

1. I think there is still room to improve the manuscript, particularly in adhering to the good practice recommendations outlined by Olofsson et al (2014) for producing scientifically rigorous and transparent estimates of accuracy and area in land use/cover change studies. For instance, the sampling design used for the study is still unclear (e.g., did you adopt a simple random sampling, or stratified random sampling, or another sampling approach for the accuracy assessment?). 

Response: Page 9, line 269-273/ Page 12&13, line 360-370_We have included the method for sampling design sued in the study. 

2. What target accuracy (or accuracies) were you aiming for in the land cover maps based on your study objectives? Based on your chosen sampling design, what was the required sample size?

Response: Page 13, line 367_We reported targeted accuracy and required sample size here. 

3. How many training and testing samples were used? 

Response: Page 9, line 269-273_We now included the method and number of training and testing samples used in the study. 

4. For the results of the accuracy assessments, you’ve responded that you have reported the accuracies (overall, user’s, and producer’s) with confidence intervals, but I do not see these reflected in Tables 1 and 2 (in commission/omission errors) or in L 357-377 in Accuracy Assessment section of Results. 

Response: Page 12, Line 361: The Confidence Interval (CI) for both years is 0.95.

5. Also, the error matrices should be presented in terms of both sample counts and estimated area proportions. Currently, Tables 1 and 2 present the error matrices for land cover maps in 2010 and 2012 in terms of sample counts, but I do not see the error matrices presented yet in terms of estimated area proportions.

Response: Page 14&15. Line 395-403_we reported both sample counts and estimated area proportion. 

6. Finally, Table 3 should then present the adjusted area estimates based on the results of the accuracy assessment.

Response: Adjusted area estimates are now included in table 3-6.

---

## [Editor Report · Decision Letter 2]

4 Aug 2020

Land Use and Land Cover Changes along the China-Myanmar Oil and Gas Pipelines – Monitoring Infrastructure Development in Remote Conflict-prone Regions

PONE-D-20-08091R2

Dear Dr. Aung,

We’re pleased to inform you that your manuscript has been judged scientifically suitable for publication and will be formally accepted for publication once it meets all outstanding technical requirements.

Kind regards,

Stephen P. Aldrich, PhD

Academic Editor

PLOS ONE
---

## [Editor Report · Acceptance letter]

6 Aug 2020

PONE-D-20-08091R2 

Land Use and Land Cover Changes along the China-Myanmar Oil and Gas Pipelines – Monitoring Infrastructure Development in Remote Conflict-prone Regions 

Dear Dr. Aung:

I'm pleased to inform you that your manuscript has been deemed suitable for publication in PLOS ONE. Congratulations! Your manuscript is now with our production department. 

Kind regards, 

on behalf of

Dr. Stephen P. Aldrich 

Academic Editor

PLOS ONE